# EvoC2F: Compiling Tool Orchestration for Efficient and Evolvable LLM Agents

Lei Wei [* † 1 2]   Qi Liu [* 2]   Ruiyang Huang [* 2]   Xiao Peng [1]   Sicong Xie [1]   Lanbo Lin [1]   Chenhao Jiang [1]
Yuanwu Xu [1]   Tianyuan Yang [1]   Jiayao Liu [1]   Li Cai [1]   Zhaolu Kang [2]   Bin Wang [2]

## Abstract

Tool-augmented language model agents have shown great potential in solving complex real-world tasks, but a key challenge remains balancing planning flexibility with the reliability required for production deployment. Existing approaches either execute tools sequentially without parallelism or generate unconstrained code, hindering optimization and verification. Additionally, agents that learn from experience often suffer from skill library pollution, where unverified abstractions degrade performance over time. We propose EvoC2F, a framework that redefines tool orchestration through program compilation and verified continuous learning. By constraining plan generation to a well-defined intermediate representation with explicit semantic annotations, EvoC2F enables provably correct optimizations, parallelism, and fault tolerance, while ensuring soundness guarantees. Our verification-gated code-to-function evolution process ensures that learned skills undergo rigorous testing before library admission. Experiments across diverse benchmarks demonstrate that EvoC2F outperforms existing methods, reducing latency and establishing a robust foundation for building reliable, evolving autonomous agents.

## 1. Introduction

The rapid advancement of Large Language Models (LLMs) has catalyzed a paradigm shift in autonomous agent development, enabling systems that interact with external tools and APIs to accomplish complex real-world tasks (Schick et al., 2023; Qin et al., 2023; Patil et al., 2024; Shen et al., 2025; Strehlow et al., 2026).

However, a fundamental tension persists between the flexibility of natural language-driven planning and the stringent reliability requirements of production deployments—current systems often generate arbitrary code or unstructured action sequences that resist systematic optimization, verification, and governance. Contemporary approaches to tool orchestration fall into two predominant categories, each with critical limitations. *Reactive execution frameworks* (Yao et al., 2022; Shinn et al., 2023; Xu et al., 2023) process tool calls sequentially without holistic planning, failing to exploit parallelization opportunities. *Code generation paradigms* (Wang et al., 2024; Huang et al., 2022; Dong et al., 2025; Li et al., 2025) produce flexible programs but lack formal semantic guarantees—without explicit dependency and side-effect annotations, automated reasoning and systematic optimization become intractable. Additionally, recent work on skill libraries (Wang et al., 2023; Ye et al., 2023; Wang et al., 2025) typically lacks rigorous verification pipelines, where unverified abstractions can corrupt planning quality over time.

To address these challenges, we propose **EvoC2F** (**Evo**lving **C**ompilable **C**ode **F**ramework), a framework that reconceptualizes tool orchestration through the lens of program compilation and verified continuous learning. Our key insight is that constraining plan generation to a well-defined intermediate representation (Plan IR) with explicit dependency and side-effect semantics enables a semantic compiler to perform provably correct optimizations—parallelization, critical path reduction, and fault tolerance injection—while maintaining formal guarantees about execution behavior.

EvoC2F operates through two tightly coupled loops. The *online execution loop* compiles and executes plans with maximal efficiency under resource constraints, constructing a directed acyclic graph (DAG) from Plan IR and annotating nodes with effect types, resource dependencies, retry policies, and idempotency requirements. The compiler optimizes this DAG to minimize makespan while respecting concurrency budgets and reliability constraints. The *offline learning loop* analyzes successful trajectories to abstract candidate macro-skills, where candidates must pass automated unit tests, contract checks, and regression evaluations before promotion to the skill library, ensuring controlled capability growth.

---
[*]Equal contribution  [†]Work done during internship at Alibaba International Digital Commerce Group.  [1]Alibaba International Digital Commerce Group [2]Peking University. Correspondence to: Xiao Peng <xinglun.px@alibaba-inc.com>.

*Proceedings of the 43rd International Conference on Machine Learning*, Seoul, South Korea. PMLR 306, 2026. Copyright 2026 by the author(s).

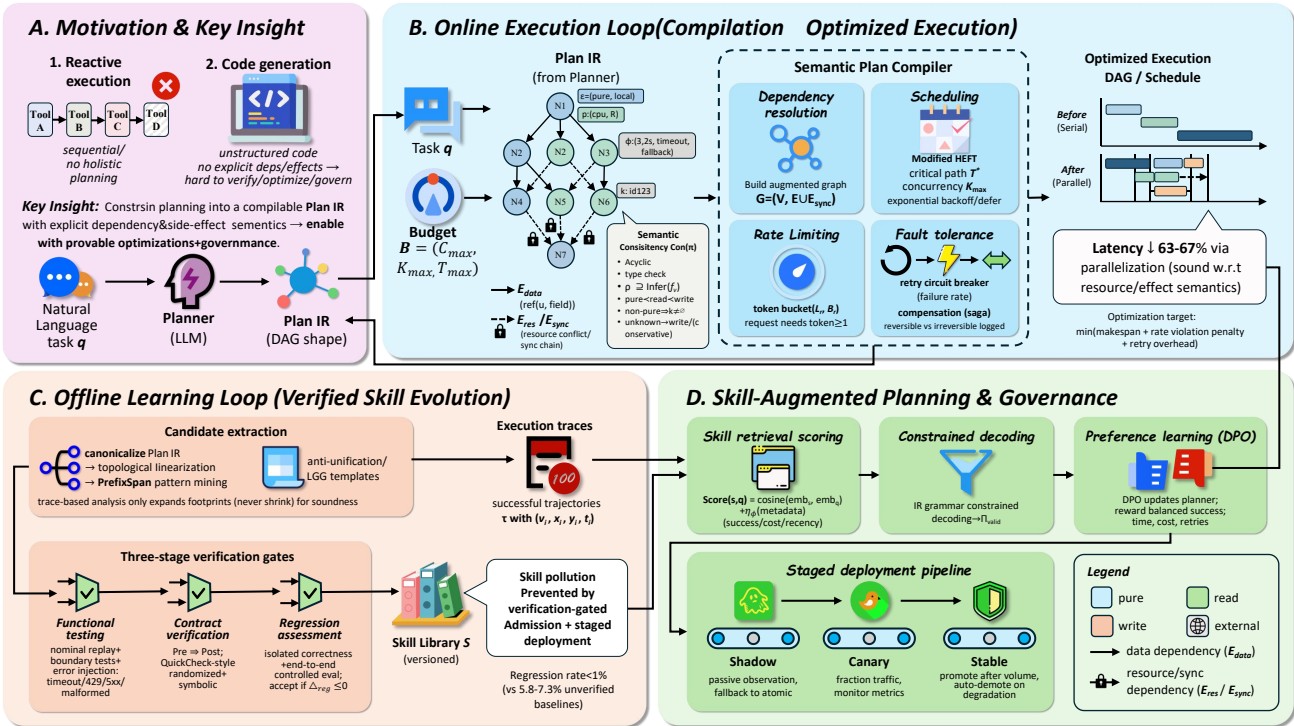

*Figure 1.* Overview of the EvoC2F framework, illustrating the compilation-based tool orchestration and verification-gated skill evolution process.

The contributions of this work are summarized as follows:

- We demonstrate that tool orchestration can be formulated as a constrained compilation problem, where explicit semantic annotations on plans enable provably correct optimizations while preventing the verification and governance challenges inherent in arbitrary code generation.

- We introduce (1) a compilable Plan IR with explicit side-effect and resource semantics enabling static analysis and optimization, (2) a semantic compiler achieving 63–67% latency reduction through effect-aware parallelization with formal correctness guarantees, and (3) a verification-gated skill evolution mechanism maintaining regression rates below 1% through automated testing, contract validation, and staged deployment.

- We present comprehensive experiments across several benchmarks demonstrating state-of-the-art success rates with substantial efficiency gains. Ablation studies validate each architectural component's necessity, and sequential evaluation over 500 tasks confirms sustained capability growth through verified skill accumulation.

## 2. Related Work

### 2.1. Tool Orchestration Frameworks

Tool orchestration for task-oriented LMs commonly spans browser-grounded acting, modular tool routing, multi-agent collaboration, and agent benchmarking (Nakano et al., 2021; Shen et al., 2023; Wu et al., 2024; Li et al., 2023a; Zhou et al., 2023; Liu et al., 2023; Li et al., 2023b). These frameworks improve coverage and scalability, but often rely on heuristic scheduling and implicit effect handling, making systematic parallelization and reliability control difficult for long-horizon tasks. EvoC2F distinguishes itself by incorporating a compilation-based approach that guarantees efficient execution through formal optimizations and parallelism, ensuring reliability without sacrificing performance.

### 2.2. Code Generation and Semantic Guarantees

Tool-mediated execution is frequently realized by synthesizing executable structures such as program-aided reasoning, computation traces, and compiled LM pipelines (Gao et al., 2023; Chen et al., 2022; Yao et al., 2023; Besta et al., 2024; Khattab et al., 2023). While these paradigms offer expressive control, they often lack explicit dependency and side-effect contracts as first-class semantics, which makes automated reasoning, scheduling, and governance brittle in production. EvoC2F addresses these shortcomings by intro-

ducing a Plan Intermediate Representation (Plan IR) with clear annotations for side-effects and resource dependencies, enabling provably correct optimizations and ensuring both efficiency and reliability.

### 2.3. Verified Execution and Skill Evolution

Verification-oriented evaluation has recently become central for agentic systems, especially via real-world issue benchmarks and test-driven regression protocols (Liu et al., 2026; Huang & Huang, 2025). These setups motivate gating mechanisms, but integrating verification directly into skill promotion and connecting it to a semantics-aware planner remains nontrivial. EvoC2F mitigates this by implementing a verification-gated skill evolution mechanism, ensuring that only verified skills are added to the agent's library, preventing skill pollution while enabling continuous improvement.

## 3. Methodology

We present EvoC2F, a framework that formulates tool orchestration as a constrained compilation problem with verified continuous learning. Our approach comprises three core components: (1) a formal Plan Intermediate Representation with explicit semantic annotations, (2) a semantic compiler that optimizes execution under resource and reliability constraints, and (3) a verification-gated skill evolution mechanism.

### 3.1. Problem Formulation

We consider an environment $\mathcal{E} = (\mathcal{T}, \mathcal{R})$ where $\mathcal{T} = \{t_1, \ldots, t_n\}$ denotes atomic tools and $\mathcal{R}$ represents shared resources (databases, APIs, file systems). Each tool $t \in \mathcal{T}$ is characterized by a tuple $t = \langle \sigma_t, \epsilon_t, \rho_t, \hat{\tau}_t, \hat{c}_t \rangle$ containing input-output signature $\sigma_t : \mathcal{X}_t \to \mathcal{Y}_t$, effect type $\epsilon_t \in \{\texttt{pure}, \texttt{read}, \texttt{write}\} \times \{\texttt{local}, \texttt{external}\}$, resource footprint $\rho_t = \{(r, a) \mid r \in \mathcal{R}, a \in \{\texttt{R}, \texttt{W}\}\}$ (pairs of resources and access modes derived from tool schema declarations), and expected latency and cost $\hat{\tau}_t, \hat{c}_t \in \mathbb{R}^+$.

Given a natural language task $q$ and budget constraint $\mathcal{B} = (C_{\max}, K_{\max}, T_{\max})$ specifying limits on cost, concurrency, and deadline, we seek a plan $\pi$ and schedule $S$ that solve the following optimization problem:

$$\min_{\pi, S} \quad \mathbb{E}_\xi[T_{\text{ms}}(S, \xi)] + \lambda_1 \Phi_{\text{rate}}(S) + \lambda_2 \Phi_{\text{retry}}(S)$$
$$\text{s.t.} \quad \sum_{v \in \pi} \hat{c}_v \leq C_{\max}, \quad \text{conc}(S) \leq K_{\max} \tag{1}$$

where the expectation is taken over random factors $\xi$ including tool latency variation, failure events, and retry counts; $T_{\text{ms}}(S, \xi)$ denotes makespan (total execution time); $\Phi_{\text{rate}}(S) = \sum_r [\text{Rate}_r(S) - L_r]_+^2$ penalizes rate limit violations, with $\text{Rate}_r(S)$ measuring the peak request rate to resource $r$ over a sliding time window and $L_r$ denoting the

rate limit; and $\Phi_{\text{retry}}(S) = \sum_v \mathbb{E}[p_{\text{fail}}(v)] \cdot n_{\text{retry}}(v) \cdot \hat{\tau}_v$ captures expected retry overhead in time units, where $p_{\text{fail}}(v)$ is the empirical failure probability and $n_{\text{retry}}(v)$ represents the expected number of retries under the configured retry policy (approximated using the maximum retry budget scaled by failure probability). The plan must additionally satisfy semantic consistency constraints detailed in Section 3.2.

### 3.2. Plan Intermediate Representation

Unlike approaches that generate arbitrary code, EvoC2F produces plans in a constrained intermediate representation amenable to static analysis and optimization.

**Definition 3.1** (Plan IR). A Plan IR is a directed acyclic graph $\pi = (V, E, \mathcal{C})$ where each node $v \in V$ represents a computational unit with attributes:

$$v = \langle f_v, \theta_v, \epsilon_v, \rho_v, \phi_v, \kappa_v \rangle \tag{2}$$

Here $f_v \in \mathcal{T} \cup \mathcal{S}$ identifies a tool or learned skill, $\theta_v$ specifies parameters potentially referencing upstream outputs via $\texttt{ref}(u, \texttt{field})$, $\epsilon_v = (e_{\text{se}}, e_{\text{env}}) \in \{\texttt{pure}, \texttt{read}, \texttt{write}\} \times \{\texttt{local}, \texttt{external}\}$ declares the effect type along two orthogonal dimensions (side-effect and environment), $\rho_v = \{(r, a) \mid r \in \mathcal{R}, a \in \{\texttt{R}, \texttt{W}\}\}$ enumerates resource accesses, $\phi_v = (n_{\max}, \gamma, \mathcal{E}_{\text{retry}}, f_{\text{fb}})$ encodes retry policy, and $\kappa_v$ provides idempotency key generation for non-pure effects.

The edge set decomposes as $E = E_{\text{data}} \cup E_{\text{res}}$, capturing distinct dependency types. Data dependencies $E_{\text{data}} = \{(u, v) \mid \theta_v \text{ references } u\}$ encode explicit information flow. To construct resource dependencies, we first establish a per-resource ordering $\prec_r$ for each resource $r \in \mathcal{R}$ by computing a topological order of the data-dependency graph $(V, E_{\text{data}})$ with deterministic tie-breaking (e.g., stable hash on node identifiers). Resource dependencies arise from potential read-write and write-read conflicts on shared state:

$$E_{\text{res}} = \{(u, v) \mid \exists r : (r, a_u) \in \rho_u \wedge (r, a_v) \in \rho_v$$
$$\wedge (a_u \neq a_v \wedge (a_u = \texttt{W} \vee a_v = \texttt{W})) \wedge u \prec_r v\} \tag{3}$$

This formulation serializes read-write and write-read conflicts while permitting concurrent read-read access. Write-write conflicts are additionally enforced through synchronization edges $E_{\text{sync}}$ introduced during compilation (Section 3.3), ensuring a complete serialization chain for all writes to each resource.

**Annotation Inference.** The resource footprint $\rho_v$ and effect type $\epsilon_v$ are derived from tool schema declarations and wrapper specifications. We define $\text{Infer}(f_v)$ as the union of all resource accesses declared in the schema or wrapper metadata for tool/skill $f_v$. For tools with incomplete or uncertain annotations, we apply a conservative

policy: unknown side-effects default to `write`, and unknown environment defaults to `external`, ensuring that under-specified tools are serialized rather than incorrectly parallelized. Trace-based analysis of historical executions is used only to monotonically expand (never shrink) the declared footprints, maintaining soundness. Runtime guards detect and log any undeclared resource accesses for future refinement.

**Definition 3.2** (Semantic Consistency). A plan $\pi = (V, E, \mathcal{C})$ is semantically consistent, denoted $\mathsf{Con}(\pi)$, iff: (i) $(V, E)$ is acyclic; (ii) $\forall (u, v) \in E_{\text{data}} : \text{type}(u.\text{out}) \preceq \text{type}(v.\text{in})$; (iii) $\forall v : \rho_v \supseteq \text{Infer}(f_v)$; (iv) side-effects respect the lattice `pure` $\prec$ `read` $\prec$ `write`; (v) $\forall v : e_{\text{se}}(v) \neq \text{pure} \Rightarrow \kappa_v \neq \varnothing$.

### 3.3. Semantic Plan Compiler

The compiler transforms semantically consistent Plan IR into optimized execution schedules through a two-phase process: compile-time dependency resolution and runtime resource coordination.

**Compile-Time Scheduling.** We first construct the augmented dependency graph $G = (V, E \cup E_{\text{sync}})$. For each resource $r \in \mathcal{R}$, let $V_r^W = \{v \in V \mid (r, \mathtt{W}) \in \rho_v\}$ denote nodes with write access. We compute a per-resource serial chain by ordering $V_r^W$ according to the same topological order used to establish $\prec_r$ (i.e., on $(V, E_{\text{data}})$), then adding synchronization edges $E_{\text{sync}}^r$ to enforce this chain. The combined synchronization edges $E_{\text{sync}} = \bigcup_r E_{\text{sync}}^r$ do not introduce cycles since they respect the underlying data-dependency order.

Let $s_v \in \mathbb{R}^+$ denote the scheduled start time of node $v$. The earliest start time (EST) and latest start time (LST) are computed via forward and backward passes:

$$\text{EST}(v) = \max_{u \in \text{pred}(v)} \left( \text{EST}(u) + \hat{\tau}_u \right) \quad (4)$$

$$\text{LST}(v) = \min_{w \in \text{succ}(v)} \left( \text{LST}(w) - \hat{\tau}_v \right) \quad (5)$$

with boundary conditions $\text{EST}(v) = 0$ for source nodes and $\text{LST}(v) = T^* - \hat{\tau}_v$ for sink nodes, where $T^* = \max_{v \in V_{\text{sink}}} (\text{EST}(v) + \hat{\tau}_v)$ is the critical path length. Nodes with positive slack $\Delta_v = \text{LST}(v) - \text{EST}(v)$ admit scheduling flexibility.

Since DAG scheduling under resource and concurrency constraints is NP-hard in general, we employ a modified HEFT (Heterogeneous Earliest Finish Time) heuristic (Topcuoglu et al., 2002). Specifically: (1) nodes are prioritized by upward rank (sum of execution time along the longest path to any sink); (2) each node is greedily assigned to the earliest feasible start time that respects all dependency edges in $E \cup E_{\text{sync}}$, the concurrency limit $K_{\max}$, and resource lock

availability; (3) rate limit constraints are checked via token bucket availability before scheduling; (4) if no feasible slot exists within the deadline, the node is deferred with exponential backoff.

**Runtime Resource Coordination.** For nodes accessing multiple resources, we employ lock ordering to prevent deadlocks: resources are assigned global identifiers, and a node must acquire locks in ascending order before execution. If acquisition fails within a timeout, the node releases held locks and retries with exponential backoff.

**Rate Limiting.** For each external resource $r$ with rate limit $L_r$ (requests per unit time), we instantiate a token bucket regulator with capacity $B_r$:

$$\text{Tokens}_r(t) = \text{clip}_{[0, B_r]} \left( \text{Tokens}_r(0) + L_r \cdot t - N_r(t) \right) \quad (6)$$

where $N_r(t)$ counts requests issued by time $t$. A request proceeds only if $\text{Tokens}_r \geq 1$, whereupon one token is consumed. The penalty term $\Phi_{\text{rate}}$ in Equation 1 provides learning-time guidance to avoid rate-limit pressure, while token buckets enforce hard limits during execution.

**Fault Tolerance.** Circuit breakers monitor failure statistics within a sliding window and halt invocations when the empirical failure rate $\hat{p}_{\text{fail}}$ exceeds service-specific tolerance, preventing cascade failures. For write operations with reversible semantics (e.g., APIs providing explicit undo endpoints), the compiler generates compensation actions $\bar{v}$ following the saga pattern (Garcia-Molina & Salem, 1987). We distinguish: (i) *reversible writes* with compensation $\bar{v}$ satisfying $\text{exec}(\bar{v}, \text{exec}(v, \sigma)) \approx \sigma$, and (ii) *irreversible external effects* (e.g., sending emails, financial transactions) which are logged for manual intervention but cannot be automatically rolled back.

**Assumption 3.3** (Annotation Soundness). For all nodes $v \in V$, the declared resource footprint $\rho_v$ is a superset of actual resources accessed during execution, and the declared effect type $\epsilon_v$ is an upper bound under the respective lattice orderings.

**Proposition 3.4** (Parallelization Soundness). *Given Assumption 3.3 and a semantically consistent plan $\pi$ with $\mathsf{Con}(\pi)$, let $S^*$ be the schedule produced by the compiler. For any nodes $u, v$ with overlapping execution intervals under $S^*$, either $\rho_u \cap \rho_v = \varnothing$, or $\forall r \in \rho_u \cap \rho_v : (r, \mathbb{R}) \in \rho_u \wedge (r, \mathbb{R}) \in \rho_v$.*

*Proof.* Read-write and write-read conflicts on any shared resource $r$ are serialized by $E_{\text{res}}$ as defined in Equation 3. Write-write conflicts are additionally serialized by the per-resource chain $E_{\text{sync}}^r$ (which totally orders $V_r^W$). The scheduler respects all edges in $E \cup E_{\text{sync}}$, enforcing $s_v \geq s_u + \hat{\tau}_u$

for $(u, v) \in E \cup E_{\text{sync}}$. By Assumption 3.3, actual resource accesses are contained within declared footprints. Thus concurrent execution under $S^*$ implies no conflicting access. $\qquad \square \qquad \square$

## 3.4. Skill-Augmented Planning

The planner generates Plan IR by leveraging both atomic tools $\mathcal{T}$ and learned skills from a dynamically growing library $\mathcal{S}$. Given task $q$, we first retrieve relevant skills by ranking candidates according to:

$$\text{Score}(s, q) = \underbrace{\cos(\mathbf{e}_s, \mathbf{e}_q)}_{\text{semantic}} + \underbrace{\eta_\phi(s, q)}_{\text{learned}} \qquad (7)$$

where $\mathbf{e}_s, \mathbf{e}_q \in \mathbb{R}^d$ are embedding representations (obtained by encoding textual descriptions of skills and tasks) and $\eta_\phi : \mathcal{S} \times \mathcal{Q} \to \mathbb{R}$ is a lightweight MLP that ingests skill metadata (historical success rate, average cost, recency) to produce a learned adjustment. The top-$k$ skills, along with tool schemas, form the augmented context $\mathcal{C}_q$ for plan generation.

The planner $\mathcal{M}_\theta$ generates Plan IR autoregressively via constrained decoding that enforces the IR grammar:

$$\pi^* = \arg\max_{\pi \in \Pi_{\text{valid}}} P_\theta(\pi \mid \mathcal{C}_q) \qquad (8)$$

where $\Pi_{\text{valid}}$ denotes the set of syntactically and semantically consistent plans.

To improve planning quality over time, we apply offline preference learning. For each completed task, trajectories are scored by a reward combining success, efficiency, and reliability:

$$R(\tau) = \mathbf{1}[\text{succ}] - \alpha_T \frac{T(\tau)}{T_{\max}} - \alpha_C \frac{C(\tau)}{C_{\max}} - \alpha_R \frac{N_{\text{retry}}}{N_{\text{budget}}} \qquad (9)$$

Given preference pairs $(\tau^+, \tau^-)$ with $R(\tau^+) > R(\tau^-)$, we update the planner via Direct Preference Optimization (Rafailov et al., 2023):

$$\mathcal{L}_{\text{DPO}} = -\mathbb{E} \left[ \log \sigma \left( \beta \log \frac{P_\theta(\pi^+ \mid \mathcal{C}_q)}{P_{\text{ref}}(\pi^+ \mid \mathcal{C}_q)} \right. \right. \\ \left. \left. - \beta \log \frac{P_\theta(\pi^- \mid \mathcal{C}_q)}{P_{\text{ref}}(\pi^- \mid \mathcal{C}_q)} \right) \right] \qquad (10)$$

where $P_{\text{ref}}$ is the frozen planner checkpoint from the previous training iteration. The retrieval router $\eta_\phi$ is jointly trained with a margin ranking objective on skill utility labels derived from trajectory outcomes (specifically, credit is assigned proportional to the performance delta when a skill is used versus when the same task is solved without it).

## 3.5. Verification-Gated Skill Evolution

The learning module continuously analyzes execution traces to extract reusable abstractions while enforcing rigorous verification to prevent skill pollution.

**Candidate Extraction.** From successful trajectories, we identify candidate macro-skills through sequential pattern mining on canonicalized Plan IR traces. We canonicalize each DAG via topological linearization with deterministic tie-breaking, then apply PrefixSpan (Pei et al., 2004) on linearized sequences. Let $\text{supp}(P) = |\{\tau \in \mathcal{D} : P \preceq \tau\}|/|\mathcal{D}|$ denote pattern support. High-support patterns with consistent data flow signatures are promoted to candidates. For structurally similar pattern families, we compute parameterized templates via anti-unification:

$$\text{LGG}(P_1, P_2) = \arg\min_{P : P_1 \unlhd P, P_2 \unlhd P} \text{Cost}(P) \qquad (11)$$

yielding the least general generalization, where $P_1 \unlhd P$ indicates $P$ generalizes $P_1$, and $\text{Cost}(P)$ measures complexity. Candidates failing $\text{Con}(\pi)$ after generalization are discarded.

**Three-Stage Verification.** Candidates enter a verification pipeline: (1) *Functional Testing* synthesizes test suites $\mathcal{T}_s = \mathcal{T}_{\text{nom}} \cup \mathcal{T}_{\text{bnd}} \cup \mathcal{T}_{\text{err}}$ covering nominal inputs, boundary conditions, and error modes; (2) *Contract Verification* checks $\forall x \in \text{Dom}(s) : \text{Pre}_s(x) \Rightarrow \text{Post}_s(s(x))$ via property-based testing and symbolic constraint solving; (3) *Regression Assessment* evaluates impact through controlled experiments:

$$\Delta_{\text{reg}}(s) = \frac{1}{|\mathcal{H}|} \sum_{q \in \mathcal{H}} \left[ \mathbf{1}[\text{fail}(q, \mathcal{S} \cup \{s\}, \xi)] - \mathbf{1}[\text{fail}(q, \mathcal{S}, \xi)] \right] \qquad (12)$$

where $\mathcal{H}$ is a held-out task set and $\xi$ controls randomness. Skills with $\Delta_{\text{reg}} \leq 0$ passing all stages are admitted.

**Staged Deployment.** Admitted skills progress through shadow, canary, and stable phases. Shadow mode enables passive observation with atomic fallback. Canary deployment routes a traffic fraction to new skills while monitoring metrics. Skills exhibiting degradation trigger automatic demotion. This lifecycle management ensures monotonic library improvement while maintaining reliability.

## 4. Experiments

### 4.1. Experimental Setup

**Datasets.** We evaluate EvoC2F on six benchmark suites: (1) StableToolBench (Guo et al., 2024) with 16,000+ real-world API calls across 49 categories for write-heavy workflows;

| Model | Method | StableToolBench | | | | ShortcutsBench | | | | | $\tau$-bench | | |
|---|---|---|---|---|---|---|---|---|---|---|---|---|---|
| | | SoPR↑ | SoWR↑ | Lat.↓ | TC↓ | ASA↑ | $Acc_{spp}$↑ | $Acc_{ofpa}$↑ | $Acc_{afni}$↑ | Lat.↓ | $pass^1_{ret}$↑ | $pass^1_{air}$↑ | Lat.↓ |
| Qwen2.5 -72B | ReAct | 56.2 | 64.8 | 18.4 | 25.7 | 39.5 | 68.4 | 82.7 | 39.1 | 15.1 | 58.6 | 31.2 | 20.3 |
| | Toolformer | 57.6 | 66.3 | 17.0 | 24.6 | 40.8 | 71.3 | 84.5 | 41.2 | 14.1 | 60.9 | 33.6 | 18.4 |
| | CodeAct | 62.1 | 69.2 | 13.3 | 21.3 | 42.9 | 73.5 | 86.2 | 41.8 | 11.0 | 66.3 | 38.4 | 15.1 |
| | Planner-Exec | 63.8 | 70.4 | 12.6 | 18.9 | 46.3 | **76.8** | 89.1 | 45.9 | 10.3 | 69.0 | 41.2 | 14.3 |
| | LLMCompiler | 60.7 | 68.1 | 7.2 | 20.5 | 44.0 | 74.6 | 87.3 | 43.5 | 6.1 | 64.5 | 36.8 | 8.5 |
| | **EvoC2F** | **65.9** | **72.5** | **6.7** | **18.4** | **50.1** | 75.9 | **91.5** | **50.6** | **5.8** | **72.7** | **44.8** | **8.2** |
| GPT-4o | ReAct | 60.4 | 68.6 | 17.6 | 24.8 | 43.6 | 73.2 | 87.1 | 44.0 | 13.9 | 64.2 | 37.6 | 19.2 |
| | Toolformer | 62.0 | 70.1 | 16.1 | 23.2 | 45.9 | 75.9 | 88.8 | 45.8 | 12.8 | 66.3 | 39.2 | 17.5 |
| | CodeAct | 66.1 | 72.9 | 12.4 | 20.1 | 47.7 | 77.9 | 90.1 | 46.3 | 10.2 | 71.5 | 43.6 | 13.8 |
| | Planner-Exec | 67.6 | 74.2 | 11.7 | **17.9** | 50.8 | 81.3 | **92.6** | 50.0 | 9.6 | 74.1 | 46.4 | 13.2 |
| | LLMCompiler | 65.2 | 72.3 | 6.4 | 19.3 | 48.9 | 79.3 | 91.2 | 48.6 | 5.6 | 70.1 | 41.6 | 7.9 |
| | **EvoC2F** | **70.6** | **77.0** | **5.9** | 18.0 | **54.7** | **84.9** | 91.8 | **53.4** | **5.3** | **78.4** | **51.2** | **7.5** |
| Claude-4 -Sonnet | ReAct | 62.3 | 70.2 | 16.9 | 24.1 | 44.8 | 74.8 | 88.3 | 45.4 | 13.2 | 65.7 | 39.2 | 18.5 |
| | Toolformer | 64.0 | 71.6 | 15.4 | 22.5 | 47.9 | 77.5 | 90.1 | 47.5 | 12.2 | 67.7 | 40.8 | 16.8 |
| | CodeAct | 67.5 | 74.1 | 11.9 | 19.4 | 49.2 | 79.1 | 90.5 | 47.1 | 9.7 | 73.6 | 45.6 | 13.3 |
| | Planner-Exec | 69.2 | 75.5 | 11.2 | 17.2 | 52.1 | 82.7 | 93.9 | 51.4 | 9.1 | 75.3 | 48.0 | 12.6 |
| | LLMCompiler | 67.0 | 73.7 | 6.1 | 18.6 | 50.2 | 80.9 | 92.5 | 49.7 | 5.3 | 72.5 | 44.4 | 7.6 |
| | **EvoC2F** | **72.2** | **78.6** | **5.5** | **16.8** | **57.0** | **86.3** | **94.7** | **56.3** | **5.0** | **79.7** | **53.6** | **7.2** |
| DeepSeek -V3.2 | ReAct | 60.5 | 69.6 | 17.3 | 24.5 | 44.3 | 73.9 | 87.6 | 44.6 | 13.6 | 65.0 | 38.4 | 18.9 |
| | Toolformer | 62.7 | 71.0 | 15.7 | 23.0 | 46.4 | 76.4 | 89.3 | 46.3 | 12.5 | 66.8 | 40.4 | 17.1 |
| | CodeAct | 66.6 | 73.5 | 12.1 | 19.8 | 48.1 | 78.6 | 89.8 | 47.2 | 9.9 | 72.7 | 45.2 | 13.5 |
| | Planner-Exec | 68.2 | 74.8 | 11.4 | 17.5 | 51.3 | 81.8 | 93.3 | **51.1** | 9.3 | 74.6 | 47.2 | 12.8 |
| | LLMCompiler | 65.9 | 73.1 | 6.3 | 19.0 | 49.4 | 80.0 | 91.7 | 49.0 | 5.5 | 71.3 | 44.0 | 7.8 |
| | **EvoC2F** | **71.4** | **76.2** | **5.7** | **17.1** | **56.3** | **85.7** | **94.1** | 50.7 | **5.2** | **79.1** | **52.8** | **7.4** |

*Table 1.* Main results across three representative benchmarks (mean over 5 runs). **StableToolBench**: SoPR (Solvable Pass Rate, %), SoWR (Solvable Win Rate vs. Llama-3.1-70B-Instruct, %), Lat. (End-to-End Latency, seconds), TC (Tool Calls per Task). **ShortcutsBench**: ASA (API Selection Accuracy, %), $Acc_{spp}$ (Static Parameter Preset Accuracy, %), $Acc_{ofpa}$ (Outputs From Previous Actions Accuracy, %), $Acc_{afni}$ (Ask-for-Input Recognition Accuracy, %), Lat. (seconds). $\tau$-**bench**: $pass^1_{ret}$ (Retail Domain Pass Rate, %), $pass^1_{air}$ (Airline Domain Pass Rate, %), Lat. (seconds). Best results in **bold**. Detailed evaluation protocols in Appendix H.

(2) ShortcutsBench (Shen et al., 2024) with Apple Short-cuts automation workflows; (3) $\tau$-bench (Yao et al., 2024) for multi-turn dialogues with policy constraints; (4) Tool-Comp (Nath et al., 2025) for dependent tool chains; (5) TRAJECT-Bench (He et al., 2025) for trajectory-level diagnostics; (6) ToolSeq-500, a sequential protocol sampling 500 tasks from StableToolBench for measuring skill accumulation dynamics. Detailed statistics are in Appendix G.1.

**Models.** We use four backbone LLMs: Qwen2.5-72B (Yang et al., 2025), GPT-4o (Achiam et al., 2023), Claude-4-Sonnet, and DeepSeek-V3.2. All models use temperature 0.2 for planning and 0 for execution. Implementation details are in Appendix G.

**Baselines.** We compare against five approaches: (1) Re-Act (Yao et al., 2022) for sequential reasoning-action; (2) Toolformer-style (Schick et al., 2023) with in-context prompting; (3) LLMCompiler (Kim et al., 2024) for parallel execution; (4) Planner-Execute (Wei et al., 2025) with DAG-based planning; (5) CodeAct (Wang et al., 2024) using Python code actions. All baselines use identical backbone models and tool access. Execution semantics are detailed in Appendix A.

**Metrics.** General metrics include: (1) Success Rate (SR) via executable checkers; (2) End-to-End Latency (mean and P95); (3) Tool Calls per Task (TC). Benchmark-specific metrics include: (1) Solvable Pass Rate (SoPR) and Solvable Win Rate (SoWR) for StableToolBench; (2) API Selection Accuracy (ASA), Static Parameter Preset Accuracy ($Acc_{spp}$), Outputs From Previous Actions Accuracy ($Acc_{ofpa}$), and Ask-for-Input Recognition Accuracy ($Acc_{afni}$) for Short-cutsBench; (3) $pass^1_{ret}$ (Retail Domain) and $pass^1_{air}$ (Airline Domain) for $\tau$-bench. All metrics are averaged over 5 runs with different random seeds.

### 4.2. Main Results

We evaluate on three representative benchmarks stressing different orchestration aspects: StableToolBench (write-heavy workflows), ShortcutsBench (workflow fidelity), and $\tau$-bench (policy compliance). Table 1 summarizes performance across all model-method combinations.

**EvoC2F achieves state-of-the-art success rates.** EvoC2F demonstrates competitive or superior performance across nearly all model-benchmark combinations. On Stable-ToolBench with Claude-4-Sonnet, EvoC2F achieves 72.2% SoPR and 78.6% SoWR, outperforming all baselines.

| Configuration | SoPR↑ | SoWR↑ | Lat.↓ | TC↓ |
|---|---|---|---|---|
| EvoC2F (Full) | **72.2** | **78.6** | **5.5** | **16.8** |
| w/o Compiler | 64.9 | 71.2 | 11.2 | 17.5 |
| w/o Effects | 64.8 | 70.9 | 6.7 | 18.9 |
| w/o Verification | 70.5 | 76.8 | 5.9 | 17.2 |
| w/o Macro-Skills | 66.1 | 72.4 | 8.1 | 21.5 |
| w/o Router | 70.8 | 77.3 | 5.8 | 17.3 |
| Planner-Exec | 69.2 | 75.5 | 11.2 | 17.2 |
| LLMCompiler | 67.0 | 73.7 | 6.1 | 18.6 |

*Table 2.* Ablation study on StableToolBench with Claude-4-Sonnet (mean over 5 runs).

| Method | SR↑ | DSR↑ | CPG↓ | Lat.↓ |
|---|---|---|---|---|
| ReAct | 62.7 | 71.3 | 3.84 | 20.9 |
| LLMCompiler | 70.8 | 78.5 | 1.92 | 12.7 |
| Planner-Execute | 74.2 | 83.7 | 1.68 | 11.3 |
| CodeAct | 71.9 | 76.2 | 2.15 | 13.5 |
| **EvoC2F** | **79.7** | **94.1** | **1.23** | **8.9** |

*Table 3.* ToolComp evaluation with GPT-4o backbone (mean over 5 runs). DSR: Dependency Satisfaction Rate (%), CPG: Critical Path Gap (ratio), Lat.: Latency (seconds).

| Method | Tool Sel.↑ | Param.↑ | Exec. Order↑ | SR↑ |
|---|---|---|---|---|
| ReAct | 78.4 | 82.1 | 85.3 | 64.2 |
| LLMCompiler | 83.2 | 85.7 | 88.9 | 71.5 |
| Planner-Execute | 86.5 | 88.3 | 91.2 | 75.8 |
| CodeAct | 81.7 | 83.5 | 83.4 | 69.3 |
| **EvoC2F** | **92.8** | **95.1** | **96.7** | **82.4** |

*Table 4.* TRAJECT-Bench trajectory quality metrics with GPT-4o (mean over 5 runs). All accuracies in percentage.

Similar trends hold with GPT-4o (70.6% SoPR, 77.0% SoWR) and DeepSeek-V3.2 (71.4% SoPR, 76.2% SoWR). With Qwen2.5-72B, EvoC2F achieves 65.9% SoPR versus Planner-Execute's 63.8% (+2.1 points), suggesting structured representations provide stronger scaffolding for less capable planners.

**Semantic compilation dramatically reduces latency.** EvoC2F reduces end-to-end latency by 63–67% compared to ReAct baselines. On StableToolBench with Claude-4-Sonnet, latency drops from 16.9s (ReAct) to 5.5s—a 67% reduction through critical path optimization and effect-aware parallelization. Similar gains are observed with GPT-4o (17.6s→5.9s, 66% reduction) and Qwen2.5-72B (18.4s→6.7s, 64% reduction). Even compared to the fastest baseline (LLMCompiler at 6.1s for Claude-4-Sonnet), EvoC2F maintains lower latency while achieving substantially higher success rates (72.2% vs 67.0% SoPR).

**Benchmark-specific metrics validate architectural contributions.** On StableToolBench, EvoC2F reduces tool calls to 16.8–18.4 per task versus 19.4–25.7 for ReAct/Toolformer/CodeAct, reflecting macro-skill abstraction reducing redundant orchestration. On ShortcutsBench with GPT-4o, EvoC2F achieves 54.7% ASA, 84.9% $Acc_{spp}$, 91.8% $Acc_{ofpa}$, and 53.4% $Acc_{afni}$, substantially exceeding CodeAct (47.7%, 77.9%, 90.1%, 46.3%), confirming that constrained IR trades flexibility for verifiable correctness. On $\tau$-bench, EvoC2F achieves 79.7% $pass^1_{ret}$ and 53.6% $pass^1_{air}$ with Claude-4-Sonnet, demonstrating compilation-time constraint injection ensures reliable governance while maintaining high task success rates across both domains.

### 4.3. Ablation Study

To quantify each component's contribution, we conduct systematic ablations on Claude-4-Sonnet across StableTool-Bench, selectively disabling: (1) **w/o Compiler**: bypass semantic compilation, execute sequentially; (2) **w/o Effects**: remove side-effect annotations, parallelize by data dependencies only; (3) **w/o Verification**: admit skills without test/regression gates; (4) **w/o Macro-Skills**: disable skill abstraction; (5) **w/o Router**: remove learned skill retrieval router $\eta_\phi$.

Table 2 shows each component contributes meaningfully. **Semantic compilation** is critical: removing it causes $2.0\times$ latency increase (11.2s vs 5.5s) and 7.3-point SoPR drop, performing similarly to Planner-Exec baseline. **Effect annotations** prevent catastrophic failures: without them, SoPR drops to 64.8% as aggressive parallelization causes resource conflicts. **Verification gates** maintain immediate performance (70.5% SoPR) but prevent long-term degradation—regression rate rises from 0.8% to 7.2% without verification. **Macro-skills** contribute 6.1 points (66.1% vs 72.2%) and reduce tool calls from 21.5 to 16.8. **Router** provides a 1.4 point gain through learned skill selection. Even the weakest variants stay within 8 points of the full system on SoPR while maintaining competitive latency, demonstrating gains arise from synergistic design rather than any single component dominating.

### 4.4. Dependency Reasoning and Trajectory Quality

To evaluate fine-grained dependency tracking and execution quality, we analyze ToolComp and TRAJECT-Bench with ground-truth dependency structures and trajectory annotations. We measure **Dependency Satisfaction Rate (DSR)** (fraction of correctly-typed parameter bindings), **Critical Path Gap (CPG)** (makespan ratio to theoretical optimum), and per-step accuracies for tool selection, parameterization, and execution ordering.

Table 3 and Table 4 demonstrate EvoC2F's superior dependency tracking: 94.1% DSR versus 83.7% for Planner-Execute, validating that explicit data-flow semantics enable

| Task Range | SR↑ | Lat.↓ | TC↓ | Skills | Reuse↑ |
|---|---|---|---|---|---|
| Tasks 1–100 | 68.2 | 6.8 | 19.2 | 23 | 12.4% |
| Tasks 101–200 | 70.4 | 6.2 | 18.1 | 51 | 28.7% |
| Tasks 201–300 | 72.0 | 5.9 | 17.4 | 78 | 41.3% |
| Tasks 301–400 | 73.4 | 5.6 | 16.9 | 103 | 49.8% |
| Tasks 401–500 | 75.0 | 5.4 | 16.5 | 134 | 58.2% |

*Skill Lifecycle (final 134 skills): 93 stable, 26 canary, 15 deprecated*
*Tasks with $\geq 2$ skills: 78.4% SR vs. 70.6% atomic-only*

*Table 5.* Sequential learning on ToolSeq-500 with Claude-4-Sonnet (mean over 5 runs). SR: Success Rate (%), Lat.: Latency (s), TC: Tool Calls per Task, Skills: Cumulative Library Size, Reuse: fraction of tasks whose compiled plan includes $\geq 1$ learned skill.

accurate parameter binding. EvoC2F achieves near-optimal parallelization (CPG=1.23) compared to sequential execution (ReAct: 3.84) and effect-agnostic approaches (LLM-Compiler: 1.92). On TRAJECT-Bench, EvoC2F achieves 92.8% tool selection, 95.1% parameterization, and 96.7% execution order accuracy, substantially exceeding code generation (CodeAct: 83.5% parameterization). Error attribution reveals EvoC2F's failures stem primarily from tool execution errors (52%) and timeouts (31%), with only 17% planning errors versus 38–47% for baselines.

## 4.5. Continual Skill Learning

We evaluate long-horizon learning with **ToolSeq-500**, a sequential protocol that processes 500 tasks sampled from our StableToolBench training pool (virtual API cache) after applying the same solvable filtering as Table 1. Tasks are processed in a fixed order, and all splits are disjoint by query ID. During execution, EvoC2F mines successful trajectories for reusable patterns and admits new skills through verification gates.

Table 5 shows **monotonic improvement** as the library grows: SR increases from 68.2% to 75.0% (+6.8), while latency drops from 6.8s to 5.4s (21%) and tool calls decrease from 19.2 to 16.5. The mining module proposes 356 candidates; verification admits 134 (37.6%), with rejections from functional tests (38%), contract violations (29%), and regression checks (33%). Regression is monitored every 20 admitted skills on a held-out validation set (100 tasks sampled from the same training pool, disjoint from ToolSeq-500), and remains below 1% throughout; without verification gates it rises to 7.2% (Table 2). Finally, skills compose: tasks invoking $\geq 2$ skills achieve 78.4% SR vs. 70.6% atomic-only, and an admitted skill encapsulates 3.7 atomic tool invocations on average.

## 4.6. Robustness and Controllability

We stress-test EvoC2F under adversarial conditions and evaluate controllability via compiler objective weights and cross-domain transfer.

| | Failure Injection (20%) | | | Strict Rate Limits (5 req/s) | | |
|---|---|---|---|---|---|---|
| Method | SR↑ | RR↓ | Lat.↓ | SR↑ | RLV↓ | TO↓ |
| ReAct | 51.2 | 2.34 | 28.7 | 48.7 | 31.4 | 23.4 |
| Planner-Exec | 62.4 | 1.78 | 18.9 | 59.8 | 22.1 | 14.2 |
| LLMCompiler | 57.8 | 2.06 | 21.3 | 52.3 | 28.7 | 19.8 |
| CodeAct | 55.3 | 2.18 | 23.1 | 54.6 | 25.8 | 17.6 |
| **EvoC2F** | **66.4** | **0.87** | **11.2** | **71.6** | **4.2** | **6.8** |

*Table 6.* Stress testing on a StableToolBench subset (200 tasks, 5 runs). SR: Success Rate (%), RR: Retry Rate, Lat.: Latency (s), RLV: Rate Limit Violations (%), TO: Timeout Rate (%).

| Configuration | SR↑ | Lat.↓ | RR↓ | RLV↓ | TC↓ |
|---|---|---|---|---|---|
| Latency-opt ($\lambda = 0$) | 68.5 | **5.1** | 0.42 | 9.3 | 17.2 |
| Balanced ($\lambda = 0.5$) | 72.4 | 6.2 | 0.23 | 4.5 | 17.8 |
| Reliable ($\lambda = 1.0$) | **74.6** | 7.8 | **0.11** | **1.8** | 18.3 |
| *Cross-Domain Transfer (StableToolBench → ShortcutsBench, metric = ASA):* | | | | | |
| Zero-shot | 49.3 | 5.8 | 0.28 | 5.7 | 15.3 |
| w/ safe fallback | 52.5 | 6.3 | 0.19 | 3.2 | 15.9 |
| Baseline (atomic) | 46.8 | 6.5 | 0.31 | 6.4 | 18.7 |

*Table 7.* Controllability and transfer learning (5 runs). Top: weight sweep on a fixed 200-task StableToolBench validation set. Bottom: transfer to ShortcutsBench reports **ASA** (teacher-forced) as the success metric; safe fallback disables 34.3% (46/134) skills using a 50-task pilot from the ShortcutsBench train split, disjoint from eval.

Table 6 shows EvoC2F remains robust under 20% failure injection (66.4% SR) and strict rate limits (71.6% SR), with lower retries (0.87), timeouts (6.8%), and violations (4.2%) than all baselines. We compute peak request-rate variance by counting per-resource requests over a 1s sliding window and taking the variance of the per-window peak rate; EvoC2F reduces this variance by $2.3\times$. For controllability, sweeping $\lambda \in \{0, 0.125, \ldots, 1.0\}$ yields a smooth trade-off (Spearman $\rho = -0.83$ using mean SR/latency over 5 runs per $\lambda$): $\lambda=0$ minimizes latency (5.1s) while $\lambda=1$ maximizes reliability (74.6% SR, 1.8% RLV). Finally, cross-domain transfer to ShortcutsBench (metric = ASA) improves from 46.8% to 49.3% zero-shot; we define negative transfer as per-task ASA dropping below atomic-only, affecting 12.3% tasks. Safe fallback disables skills that decrease pilot ASA, improving to 52.5% while suppressing 46/134 skills; a task is counted as invoking transferred skills if its compiled plan contains at least one transferred skill node (31.6% tasks).

## 5. Conclusion

EvoC2F introduces a novel approach to tool orchestration by leveraging a semantic Plan IR for efficient and reliable execution. Its use of a semantic compiler allows for sound parallelization and critical-path optimization, while the verification-gated skill evolution process ensures that new skills are rigorously tested before being added to the agent's library. The framework achieves state-of-the-art performance in various benchmarks, with substantial reductions in latency and tool calls, and demonstrates robust,

long-term skill accumulation.

## 6. Limitations

Despite its strengths, EvoC2F faces several challenges. The framework's scalability to large, distributed systems remains an open problem, and its reliance on detailed Plan IR may impose overheads that affect real-time responsiveness. Additionally, while the verification process ensures skill quality, it may limit the framework's ability to generalize learned skills across diverse tasks without further tuning or retraining.

## Impact Statement

EvoC2F targets the engineering pipeline by which LLM agents invoke real-world tools, so its societal effects are mediated by how and where those agents are deployed. On the positive side, semantic compilation and effect-aware scheduling reduce end-to-end latency by 63–67% and substantially cut redundant tool calls, lowering the compute, network, and energy cost of running agentic systems at scale. The explicit separation between reversible writes and irreversible external effects, together with verification gates on admitted skills, is intended to make autonomous tool use safer in settings where mistakes are costly: writes to shared resources are serialized rather than parallelized, irreversible actions such as financial transactions or outbound notifications are routed through manual-intervention logs rather than automatic retries, and circuit breakers halt invocation when empirical failure rates exceed service-specific tolerances.

These same properties also raise concerns. Lowering the operational cost and observable error rate of tool-using agents reduces the barrier to deploying them in high-stakes settings where human-in-the-loop oversight may still be warranted. The soundness of our parallelization argument rests on the accuracy of tool schema annotations: incorrect or incomplete annotations can lead to silent resource conflicts that the compiler will not detect, even though our conservative defaults (unknown side-effects default to `write`, unknown environment to `external`) are designed to mitigate this. The verification-gated skill library reduces but does not eliminate the risk that admitted skills encode subtle errors or biases inherited from the trajectories from which they were mined; periodic re-verification, staged rollouts, and rollback paths remain necessary in any long-running deployment. We encourage practitioners adopting EvoC2F-style compilation to treat the IR-level annotations as a safety-critical surface, and to combine them with downstream guardrails—rate-limit budgets, audit logs, and human review for high-impact actions—rather than relying on the compiler alone.

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

# A. Baseline Implementation Details and Fairness

To ensure fair comparison, all baseline methods share identical experimental infrastructure with EvoC2F across tool access, execution environment, and failure handling.

**Shared infrastructure.** All methods operate over the same tool collection $\mathcal{T}$ with identical tool schemas, sandbox environments, and failure models. Tool invocations execute in isolated containers with deterministic timeout (30s per call) and failure injection rates controlled by experimental conditions (Section 4.6). All methods have access to the same retry budgets ($n_{\max} = 3$ retries with exponential backoff base $\gamma = 2.0$), rate limit specifications (method-agnostic token bucket parameters $L_r, B_r$ per resource), and cost constraints ($C_{\max}$ set per benchmark). Prompts for all methods include identical tool documentation, input-output schemas, and task descriptions, differing only in method-specific formatting (e.g., ReAct includes reasoning traces, CodeAct wraps in Python syntax).

**ReAct and Toolformer.** We implement ReAct following Yao et al. (2022) with interleaved thought-action-observation loops. At each step, the model generates a reasoning trace followed by a single tool invocation or final answer. Observations from tool execution are appended to the prompt for the next iteration, with a maximum of 15 reasoning steps before forced termination. Toolformer follows Schick et al. (2023) with in-context demonstrations of tool usage patterns. The model generates text interspersed with tool calls in the format `[ToolName(arg1=val1, ...)]`, which are parsed and executed sequentially. Both methods execute tools strictly one-at-a-time without lookahead or parallelization.

**CodeAct.** Following Wang et al. (2024), we allow the model to generate executable Python code with tool invocations as function calls. The generated code is executed in a sandboxed Python interpreter with tool functions exposed in the global namespace. We use the publicly available implementation from the CodeAct repository[1] with default parameters: temperature 0.2 for code generation, maximum 10 code blocks per task, and 60s timeout per block execution. Unlike EvoC2F's Plan IR which enforces explicit dependency annotations, CodeAct allows arbitrary control flow including loops and conditionals, providing maximum flexibility at the cost of reduced analyzability.

**LLMCompiler.** We implement LLMCompiler following Kim et al. (2024), which generates a dependency-annotated task graph and schedules nodes for parallel execution. The planner outputs a JSON representation specifying tool calls with explicit `depends_on` fields. We use the reference implementation[2] with modifications to match our evaluation harness: concurrency limit $K_{\max}$ enforced via semaphore, and retry policies aligned with EvoC2F (same $n_{\max}, \gamma$ parameters). Critically, LLMCompiler lacks explicit side-effect annotations and performs dependency analysis solely from data-flow declarations, potentially leading to read-write conflicts when tools have undeclared side-effects.

**Planner-Execute.** We implement a planner-executor separation where the planner first generates a complete DAG of tool invocations with data dependencies, then a separate executor schedules and runs the graph. The planner uses a structured JSON schema similar to EvoC2F's Plan IR but without explicit effect types $\epsilon_v$ or resource footprints $\rho_v$. The executor performs topological scheduling with greedy parallelization subject to data dependencies and concurrency limits, but without rate-aware scheduling or effect-based conflict detection. This baseline isolates the benefit of planning from EvoC2F's semantic compilation optimizations.

**Alignment of retry and rate-limit policies.** All methods share identical retry configurations: exponential backoff with base $\gamma = 2.0$, maximum retries $n_{\max} = 3$, and retryable error classes (timeouts, 5xx responses, rate-limit 429s). For rate limiting, all methods are subject to the same token bucket constraints $(L_r, B_r)$ per resource $r$, enforced by a shared rate-limiter middleware that blocks requests when tokens are unavailable. The key difference is that EvoC2F's compiler proactively schedules to avoid rate-limit pressure (via $\Phi_{\text{rate}}$ penalty and bucket-aware timing), while baselines reactively back off upon encountering 429 responses. This difference reflects architectural capability rather than unfair advantage—baselines could in principle implement similar lookahead if equipped with explicit resource annotations.

**Skill libraries for baselines.** To ensure fair comparison in continual learning experiments (Section 4.5), we provide ReAct and CodeAct with access to accumulated execution traces and allow them to reference past successful solutions via retrieval-augmented prompting. Specifically, for each new task, we retrieve the top-3 most similar historical trajectories

---

[1] https://github.com/xingyaoww/code-act
[2] https://github.com/SqueezeAILab/LLMCompiler

(by task embedding similarity) and include them as in-context examples. This gives baselines an opportunity to reuse patterns without requiring the formal verification pipeline. LLMCompiler and Planner-Execute do not natively support skill abstraction, so we evaluate them in their standard single-task formulation.

## B. Token Bucket Constraints and Rate Penalty Consistency

We formalize the relationship between EvoC2F's soft rate-limit penalty $\Phi_{\text{rate}}(S)$ used during plan optimization (Equation 1) and the hard token bucket constraints enforced during execution. This establishes that minimizing the soft penalty during compilation produces schedules that satisfy hard rate limits at runtime.

**Token bucket model.** For each external resource $r \in \mathcal{R}$ with rate limit $L_r$ (requests per unit time) and burst capacity $B_r$, we maintain a token bucket with state $\text{Tok}_r(t) \in [0, B_r]$ denoting available tokens at time $t$. Tokens regenerate at rate $L_r$ and are consumed by requests. In discrete time with update interval $\Delta$, the dynamics are:

$$\text{Tok}_r(t + \Delta) = \min\{B_r, \text{Tok}_r(t) + L_r \cdot \Delta\} - n_r(t, t + \Delta) \tag{13}$$

where $n_r(t, t + \Delta)$ is the number of requests to resource $r$ in the interval $[t, t + \Delta)$. A request at time $t$ is admitted if $\text{Tok}_r(t) \geq 1$, upon which one token is consumed. If $\text{Tok}_r(t) < 1$, the request blocks until sufficient tokens accumulate.

**Feasibility condition.** A schedule $S$ is *rate-feasible* for resource $r$ if for any time window $W = [t_1, t_2]$ of length $|W| = t_2 - t_1$, the number of requests satisfies:

$$N_r(W) \leq L_r \cdot |W| + B_r \tag{14}$$

This is the classical token bucket feasibility condition: over any window, request count cannot exceed the base rate multiplied by window duration plus the burst allowance. When this condition holds and the bucket starts with sufficient initial tokens (typically $\text{Tok}_r(0) = B_r$), the token count never becomes negative, ensuring all requests are admitted without blocking.

**Peak rate and soft penalty.** EvoC2F's compiler computes the peak request rate over sliding windows during schedule construction. Specifically, for schedule $S$ and resource $r$, we define $\text{Rate}_r(S)$ as:

$$\text{Rate}_r(S) = \max_{\substack{W = [t_1, t_2] \\ |W| \geq \Delta_{\min}}} \frac{N_r(W)}{|W|} \tag{15}$$

where $\Delta_{\min}$ is a minimum window size (set to $1/L_r$ to capture instantaneous bursts) and $N_r(W)$ counts nodes in $S$ accessing $r$ with execution intervals overlapping $W$. The soft penalty in Equation 1 is:

$$\Phi_{\text{rate}}(S) = \sum_{r \in \mathcal{R}} \left[\text{Rate}_r(S) - L_r\right]_+^2 \tag{16}$$

where $[\cdot]_+$ denotes the positive part. This penalty is zero when the peak rate over any window does not exceed $L_r$, and grows quadratically when violations occur.

**Consistency between soft penalty and hard constraint.** We now establish that schedules minimizing $\Phi_{\text{rate}}$ respect token bucket feasibility. Suppose $\Phi_{\text{rate}}(S) = 0$, which implies $\text{Rate}_r(S) \leq L_r$ for all $r$. Then for any window $W$ of length $|W|$:

$$N_r(W) \leq \text{Rate}_r(S) \cdot |W| \leq L_r \cdot |W| \leq L_r \cdot |W| + B_r \tag{17}$$

where the final inequality holds since $B_r \geq 0$. Thus $S$ satisfies the feasibility condition (Equation 11) for all resources. Conversely, if $\Phi_{\text{rate}}(S) > 0$, there exist a resource $r$ and a witness window $W^*$ with $\delta_0 := \text{Rate}_r(S) - L_r > 0$ and $N_r(W^*) \geq (L_r + \delta_0) \cdot |W^*|$. Whenever the witness window satisfies $|W^*| > B_r/\delta_0$, we obtain $N_r(W^*) > L_r \cdot |W^*| + B_r$, violating the feasibility condition (Equation 11).

This establishes a practical guarantee: during compilation, EvoC2F's optimizer (which includes $\lambda_1 \Phi_{\text{rate}}$ in its objective) is incentivized to produce schedules with $\text{Rate}_r(S) \leq L_r$, ensuring that when these schedules execute under token bucket regulation at runtime, requests are admitted without blocking (up to stochastic latency variation and minor timing jitter). The soft penalty thus serves as a differentiable surrogate for the hard bucket constraint, enabling gradient-based or heuristic optimization during planning while guaranteeing runtime compliance. In practice, we set $\lambda_1 > 0$ to trade off makespan minimization against rate-limit safety, producing schedules that operate below peak capacity with controlled slack (as demonstrated in Table 7).

## C. Parallelization Correctness and Deadlock Freedom

We provide formal proofs establishing three key properties of EvoC2F's semantic compiler: (1) synchronization edges $E_{\text{sync}}$ preserve acyclicity, (2) the resulting schedule respects all resource conflicts, and (3) the multi-resource locking protocol is deadlock-free.

### C.1. Acyclicity Preservation

**Theorem C.1** (Synchronization Edges Preserve DAG Structure). *Let $\pi = (V, E_{data}, \mathcal{C})$ be a semantically consistent plan with $(V, E_{data})$ acyclic, and let $E_{sync}$ be the synchronization edges constructed as in Section 3.3. Then $(V, E_{data} \cup E_{sync})$ is acyclic.*

*Proof.* Let $\prec$ denote a topological order on $(V, E_{\text{data}})$ with deterministic tie-breaking (e.g., lexicographic ordering on node identifiers). By construction, for each resource $r$ and the set of nodes with write access $V_r^W = \{v \in V \mid (r, \mathbb{W}) \in \rho_v\}$, we impose a total order $u \prec_r v$ for all $u, v \in V_r^W$ with $u \prec v$, and add synchronization edges $E_{\text{sync}}^r = \{(u, v) \mid u, v \in V_r^W, u \prec_r v, \nexists w \in V_r^W : u \prec w \prec v\}$ to enforce consecutive pairs in this chain. The combined synchronization set is $E_{\text{sync}} = \bigcup_r E_{\text{sync}}^r$.

Suppose for contradiction that $G' = (V, E_{\text{data}} \cup E_{\text{sync}})$ contains a cycle $C = v_1 \to v_2 \to \cdots \to v_k \to v_1$. Since $(V, E_{\text{data}})$ is acyclic, at least one edge in $C$ must belong to $E_{\text{sync}}$. Let $(v_i, v_j) \in E_{\text{sync}}$ be such an edge, meaning there exists resource $r$ with $v_i, v_j \in V_r^W$ and $v_i \prec v_j$ (by construction of $E_{\text{sync}}^r$). Now consider the path $v_j \to \cdots \to v_i$ completing the cycle. This path consists of edges from $E_{\text{data}} \cup E_{\text{sync}}$.

For any edge $(u, w)$ along this path:

- If $(u, w) \in E_{\text{data}}$, then $u \prec w$ since $\prec$ is a topological order of $E_{\text{data}}$.

- If $(u, w) \in E_{\text{sync}}^{r'}$ for some resource $r'$, then by the same construction argument, $u \prec w$.

Therefore, every edge in the path $v_j \to \cdots \to v_i$ satisfies $u \prec w$, implying by transitivity that $v_j \prec v_i$. But this contradicts $v_i \prec v_j$ established earlier. Hence no such cycle can exist, and $G'$ is acyclic. □ □

This theorem guarantees that adding synchronization edges to serialize write conflicts does not introduce cyclic dependencies, preserving the schedulability of the plan.

### C.2. Parallelization Soundness

We now strengthen the parallelization soundness result from the main text (Proposition following Assumption 3.3) by explicitly addressing how resource dependencies $E_{\text{res}}$ and synchronization edges $E_{\text{sync}}$ together prevent all resource conflicts.

**Theorem C.2** (Schedule Respects Resource Conflicts). *Let $\pi = (V, E_{data}, \mathcal{C})$ be a semantically consistent plan satisfying Assumption 3.3 (annotation soundness), and let $S$ be the schedule produced by the compiler from the augmented graph $G = (V, E_{data} \cup E_{res} \cup E_{sync})$. Then for any pair of nodes $u, v \in V$ with overlapping execution intervals under $S$ (i.e., $[s_u, s_u + \tau_u] \cap [s_v, s_v + \tau_v] \neq \emptyset$), either:*

*(i) $\rho_u \cap \rho_v = \emptyset$ (disjoint resource footprints), or*

*(ii) $\forall r \in \rho_u \cap \rho_v : (r, \mathbb{R}) \in \rho_u \wedge (r, \mathbb{R}) \in \rho_v$ (both are read-only accesses).*

*Proof.* The scheduler enforces $s_w \geq s_x + \tau_x$ whenever $(x, w) \in G$ (Algorithm 7, dependency bound on line 2), so overlapping execution intervals between $u$ and $v$ imply neither is an ancestor of the other in $G$. Consider any shared resource $r \in \rho_u \cap \rho_v$.

**Case 1: At least one write.** Suppose $(r, \mathbb{W}) \in \rho_u$ (the case $(r, \mathbb{W}) \in \rho_v$ is symmetric). By the construction of $E_{\text{res}}$ (Equation 3), there exists an edge $(u, v) \in E_{\text{res}}$ if $u \prec_r v$ (where $\prec_r$ is the per-resource topological order), or $(v, u) \in E_{\text{res}}$ if $v \prec_r u$. Since $u$ has write access, $u \in V_r^W$, and by construction of $E_{\text{sync}}^r$, all writes to $r$ are totally ordered via a chain of synchronization edges. If $v$ also has write access (i.e., $v \in V_r^W$), then there exists a directed path from one to the other in

$(V, E_{\text{sync}}^r)$. If $v$ has only read access but shares resource $r$ with writer $u$, then $E_{\text{res}}$ contains an edge serializing the read-write conflict. In either case, there exists a directed path from one node to the other in $G$, contradicting the assumption that $u$ and $v$ have overlapping execution intervals (the scheduler respects all edges in $G$).

**Case 2: Both reads.** If $(r, \mathtt{R}) \in \rho_u$ and $(r, \mathtt{R}) \in \rho_v$, then no edge is introduced in $E_{\text{res}}$ or $E_{\text{sync}}$ for this resource (read-read conflicts are permitted). This matches condition (ii).

By Assumption 3.3, the declared footprints $\rho_u, \rho_v$ are supersets of actual resources accessed during execution, so any undeclared access would violate the assumption. Therefore, concurrent execution under $S$ respects all resource conflicts. □ □

**Conservative handling of unknown effects.** When a tool's side-effects are incompletely specified, EvoC2F applies a conservative default: unknown side-effects are treated as `write` and unknown environments as `external`. This ensures Assumption 3.3 holds even for under-annotated tools, at the cost of reduced parallelism (such tools are serialized with respect to all other accesses to the same resource). Over time, trace-based monitoring can refine these annotations: if a tool consistently exhibits read-only behavior in practice, its declared effect can be downgraded to `read`, monotonically expanding the set of safe parallelizations. Runtime guards detect any undeclared resource accesses (via sandbox instrumentation) and log them for future annotation updates, maintaining soundness throughout the system's lifecycle.

### C.3. Deadlock Freedom via Lock Ordering

When a node accesses multiple resources, EvoC2F must acquire locks on all required resources before execution. Naively acquiring locks in arbitrary order can lead to circular wait conditions and deadlock. We prevent this via a global resource ordering protocol.

**Theorem C.3** (Multi-Resource Lock Ordering Prevents Deadlock). *Let $\prec_{global}$ be a total order on resources $\mathcal{R}$ (e.g., by resource identifier hash). Suppose every node $v$ acquires locks on $\{r \mid (r, a) \in \rho_v\}$ in ascending order according to $\prec_{global}$, and releases all locks upon completion. Then the system is deadlock-free.*

*Proof.* We prove by contradiction using the classical Coffman conditions for deadlock, specifically showing that circular wait cannot occur. Suppose a deadlock exists, meaning there is a set of nodes $\{v_1, \ldots, v_k\}$ such that each $v_i$ holds lock $r_i$ and waits for lock $r_{i+1}$ (with $v_{k+1} = v_1$, forming a cycle). By the acquisition protocol, node $v_i$ acquires locks in ascending order $\prec_{global}$. Since $v_i$ holds $r_i$ and waits for $r_{i+1}$, we must have $r_i \prec_{global} r_{i+1}$ (otherwise $v_i$ would have attempted to acquire $r_{i+1}$ before $r_i$, or would have already acquired $r_{i+1}$). Applying this reasoning around the cycle:

$$r_1 \prec_{\text{global}} r_2 \prec_{\text{global}} \cdots \prec_{\text{global}} r_k \prec_{\text{global}} r_1 \tag{18}$$

But $\prec_{\text{global}}$ is a total order, hence transitive and irreflexive, so $r_1 \prec_{\text{global}} r_1$ is impossible. This contradiction shows no such cycle can exist, and therefore no deadlock can occur. □ □

In practice, nodes implement this protocol with timeout-based lock acquisition: if a node cannot acquire all required locks within a timeout window (set to $5\times$ the expected makespan), it releases all held locks and retries with exponential backoff. This prevents livelock in the presence of transient contention. The combination of acyclic dependency graphs (Theorem C.1), conflict-respecting schedules (Theorem C.2), and deadlock-free locking (Theorem C.3) establishes the formal correctness of EvoC2F's parallel execution semantics.

## D. Plan IR Complete Specification

This appendix provides the formal specification of Plan IR, including grammar, type system, static verification procedures, and complete examples demonstrating the compilation process.

### D.1. Grammar and JSON Example

A Plan IR instance is represented as a JSON object. Listing 1 shows an illustrative example of a single node (with the `$ref` marker used to encode upstream references inside a parameter value) and one data edge; field types and admissible values are specified in the type system below.

*Listing 1.* Plan IR JSON example (illustrative; not a JSON Schema document).

```json
{
  "nodes": [
    {
      "id": "string",
      "function": "string",
      "parameters": {
        "param_name": "value_or_ref",
        "$ref": ["node_id", "field_path"]
      },
      "effect": {
        "side_effect": "read",
        "environment": "local"
      },
      "resources": [
        {"resource": "string", "access": "R"}
      ],
      "retry_policy": {
        "max_retries": 3,
        "backoff_gamma": 2.0,
        "retry_on": ["error_class"],
        "fallback": null
      },
      "idempotency_key": null,
      "output_schema": "type_expr",
      "latency_estimate": 0.0,
      "cost_estimate": 0.0
    }
  ],
  "edges": [
    {"from": "string", "to": "string", "type": "data"}
  ]
}
```

**Reference Syntax.**  Parameter values may reference upstream outputs via the $ref construct:

$$\text{ref}(u, p) \triangleq \{\texttt{"\$ref"} : [u.\text{id}, p]\} \tag{19}$$

where $p$ is a field path string (e.g., `"result.items[0].id"`) specifying JSONPath-style extraction from node $u$'s output. The path $p$ is parsed into accessor sequence $p = [a_1, a_2, \ldots, a_k]$ where each $a_i$ is either a field name or array index.

**Effect Lattice.**  Side-effect types form a lattice $(\mathcal{E}_{\text{se}}, \preceq_{\text{se}})$ with:

$$\texttt{pure} \prec_{\text{se}} \texttt{read} \prec_{\text{se}} \texttt{write} \tag{20}$$

Environment types form a lattice $(\mathcal{E}_{\text{env}}, \preceq_{\text{env}})$ with:

$$\texttt{local} \prec_{\text{env}} \texttt{external} \tag{21}$$

The combined effect type is the product lattice $\epsilon_v = (e_{\text{se}}, e_{\text{env}}) \in \mathcal{E}_{\text{se}} \times \mathcal{E}_{\text{env}}$ with pointwise ordering:

$$(e_1, e_1') \preceq (e_2, e_2') \iff e_1 \preceq_{\text{se}} e_2 \wedge e_1' \preceq_{\text{env}} e_2' \tag{22}$$

**Resource Footprint Format.**  Each resource access is a tuple $(r, a)$ where $r \in \mathcal{R}$ is a resource identifier (URI, database name, API endpoint) and $a \in \{\texttt{R}, \texttt{W}\}$ denotes access mode. The footprint $\rho_v = \{(r_1, a_1), \ldots, (r_m, a_m)\}$ is a set with unique resource identifiers.

**Resource Footprint Normalization.**  When a resource $r$ appears multiple times in the JSON list with different access modes, we apply the normalization rule:

$$(r, \texttt{R}) \sqcup (r, \texttt{W}) = (r, \texttt{W}) \tag{23}$$

ensuring that $\rho_v$ is a proper set with unique resource identifiers, where write access dominates read access.

## D.2. Type System

**Base Types.** We define a type language $\mathcal{T}$ with base types and constructors:

$$\tau ::= \texttt{Int} \mid \texttt{Float} \mid \texttt{String} \mid \texttt{Bool} \mid \texttt{Any} \tag{24}$$
$$\mid \texttt{Record}(\{\ell_1 : \tau_1, \ldots, \ell_n : \tau_n\}) \tag{25}$$
$$\mid \texttt{Optional}(\tau) \tag{26}$$
$$\mid \texttt{List}(\tau) \tag{27}$$
$$\mid \tau_1 \cup \tau_2 \quad \text{(union type)} \tag{28}$$

**Subtyping Relation.** The subtyping relation $\tau_1 \preceq \tau_2$ (read "$\tau_1$ is a subtype of $\tau_2$") is defined inductively:

(i) **Reflexivity:** $\tau \preceq \tau$ for all $\tau \in \mathcal{T}$.

(ii) **Top type:** $\tau \preceq \texttt{Any}$ for all $\tau$.

(iii) **Record depth subtyping:**

$$\texttt{Record}(\{\ell_1 : \tau_1, \ldots, \ell_n : \tau_n, \ell_{n+1} : \tau_{n+1}, \ldots\}) \preceq$$
$$\texttt{Record}(\{\ell_1 : \tau_1', \ldots, \ell_n : \tau_n'\})$$
$$\text{if } \tau_i \preceq \tau_i' \text{ for all } i \in [1, n] \tag{29}$$

(iv) **Optional covariance:** $\texttt{Optional}(\tau_1) \preceq \texttt{Optional}(\tau_2)$ if $\tau_1 \preceq \tau_2$.

(v) **List covariance:** $\texttt{List}(\tau_1) \preceq \texttt{List}(\tau_2)$ if $\tau_1 \preceq \tau_2$.

(vi) **Union subtyping:** $\tau \preceq \tau_1 \cup \tau_2$ if $\tau \preceq \tau_1$ or $\tau \preceq \tau_2$.

(vii) **Transitivity:** If $\tau_1 \preceq \tau_2$ and $\tau_2 \preceq \tau_3$, then $\tau_1 \preceq \tau_3$.

**Type Compatibility Check.** For data dependency edge $(u, v) \in E_{\text{data}}$ where parameter $\theta_v[p] = \texttt{ref}(u, q)$, we require:

$$\text{Extract}(\sigma_u.\texttt{out}, q) \preceq \sigma_v.\texttt{in}[p] \tag{30}$$

where $\text{Extract}(\tau, q)$ computes the type obtained by following path $q$ through type $\tau$:

$$\text{Extract}(\texttt{Record}(\{\ldots, \ell : \tau, \ldots\}), [\ell] \cdot q') = \text{Extract}(\tau, q') \tag{31}$$
$$\text{Extract}(\texttt{List}(\tau), [i] \cdot q') = \text{Extract}(\tau, q') \tag{32}$$
$$\text{Extract}(\texttt{Optional}(\tau), q) = \texttt{Optional}(\text{Extract}(\tau, q)) \tag{33}$$
$$\text{Extract}(\tau, []) = \tau \tag{34}$$

## D.3. Static Verification

The static checker $\textsf{Verify} : \pi \to \{\texttt{valid}, \texttt{error}\}$ performs the following checks:

**Acyclicity Check.** We use depth-first search with color marking:

$$\textsf{IsAcyclic}(G) = \neg\exists v \in V : \textsf{DFS}(v, \{\}, \{\}) = \texttt{cycle} \tag{35}$$

where $\textsf{DFS}(v, \text{gray}, \text{black})$ returns $\texttt{cycle}$ if a back edge to a gray node is encountered during traversal. Complexity: $O(|V| + |E|)$.

**Reference Validity.** For each reference $\texttt{ref}(u, q)$ in $\theta_v$:

(a) $(u, v) \in E_{\text{data}}$ must exist (enforced by DAG construction).

(b) Path $q$ must be valid for $\sigma_u.\texttt{out}$: $\text{Extract}(\sigma_u.\texttt{out}, q) \neq \bot$.

(c) No forward references: if $(u, v) \in E_{\text{data}}$, then $u$ must precede $v$ in topological order.

---

**Algorithm 1** Static Verification

---

1: **Input:** $\pi = (V, E, \mathcal{C})$
2: $G \leftarrow (V, E)$
3: **if** $\neg \mathsf{IsAcyclic}(G)$ **then**
4:     **return** `error("Cycle detected")`
5: **end if**
6: **for all** $(u, v) \in E_{\mathrm{data}}$ **do**
7:     **for all** $p \in \mathrm{dom}(\theta_v)$ where $\theta_v[p] = \mathtt{ref}(u, q)$ **do**
8:         $\tau_{\mathrm{out}} \leftarrow \mathrm{Extract}(\sigma_u.\mathtt{out}, q)$
9:         $\tau_{\mathrm{in}} \leftarrow \sigma_v.\mathtt{in}[p]$
10:         **if** $\neg(\tau_{\mathrm{out}} \preceq \tau_{\mathrm{in}})$ **then**
11:             **return** `error("Type mismatch at edge (u, v)")`
12:         **end if**
13:     **end for**
14: **end for**
15: **for all** $v \in V$ **do**
16:     $\rho_v^{\mathrm{inferred}} \leftarrow \mathrm{Infer}(f_v)$
17:     **if** $\neg(\rho_v^{\mathrm{inferred}} \subseteq \rho_v)$ **then**
18:         **return** `error("Incomplete resource annotation at v")`
19:     **end if**
20:     **if** $e_{\mathrm{se}}(v) = \mathtt{write} \wedge \kappa_v = \mathtt{null}$ **then**
21:         **return** `error("Missing idempotency key at v")`
22:     **end if**
23: **end for**
24: **return** `valid`

---

**Idempotency Key Generation.** For nodes with $e_{\mathrm{se}}(v) = \mathtt{write}$, the idempotency key $\kappa_v$ is generated as:

$$\kappa_v = \mathrm{Hash}(f_v \| \mathrm{Canonicalize}(\theta_v) \| \mathrm{nonce}) \tag{36}$$

where Canonicalize produces deterministic parameter encoding and nonce ensures uniqueness across plan instances. For pure nodes and read-only operations, $\kappa_v = \mathtt{null}$. Note that for external read operations that require exactly-once semantics (e.g., metered APIs), the system can optionally generate idempotency keys by setting the effect to $(\mathtt{read}, \mathtt{external})$ with explicit key generation.

**Conservative Annotation Policy.** When tool metadata is incomplete or uncertain, we apply upper-bound inference:

$$\mathrm{Infer}_{\mathrm{conservative}}(f_v) = \begin{cases} \rho_v^{\mathrm{declared}} \cup \rho_v^{\mathrm{trace}} & \text{if metadata available} \\ \{(\texttt{"unknown\_external"}, \mathtt{W})\} & \text{otherwise} \end{cases} \tag{37}$$

$$e_{\mathrm{se}}^{\mathrm{conservative}}(v) = \max_{\preceq_{\mathrm{se}}} \{e_{\mathrm{se}}^{\mathrm{declared}}, e_{\mathrm{se}}^{\mathrm{trace}}\} \tag{38}$$

$$e_{\mathrm{env}}^{\mathrm{conservative}}(v) = \max_{\preceq_{\mathrm{env}}} \{e_{\mathrm{env}}^{\mathrm{declared}}, e_{\mathrm{env}}^{\mathrm{trace}}\} \tag{39}$$

where $\rho_v^{\mathrm{trace}}$ and $e^{\mathrm{trace}}$ are extracted from historical execution traces via runtime monitoring, and the special resource identifier `"unknown_external"` represents global external state, ensuring serialization with all external write operations while remaining computable. This policy ensures soundness: over-approximating effects may reduce parallelism but never causes incorrect concurrent execution.

## E. Semantic Compiler Algorithms and Complexity Analysis

This appendix details the compilation pipeline, scheduling algorithms, resource coordination mechanisms, and complexity analysis.

---

**Algorithm 2** Semantic Compilation

---

1: **Input:** $\pi = (V, E, \mathcal{C}), \mathcal{B} = (C_{\max}, K_{\max}, T_{\max})$
2: *// Phase 1: Dependency Graph Construction*
3: $E_{\text{data}} \leftarrow \textsf{BuildDataDependencies}(V, E)$
4: $E_{\text{res}} \leftarrow \textsf{BuildResourceDependencies}(V, E_{\text{data}})$
5: $E_{\text{sync}} \leftarrow \bigcup_{r \in \mathcal{R}} \textsf{BuildWriteChain}(V, r, E_{\text{data}})$
6: $G \leftarrow (V, E_{\text{data}} \cup E_{\text{res}} \cup E_{\text{sync}})$
7:
8: *// Phase 2: Scheduling with Resource Constraints*
9: $\text{rank} \leftarrow \textsf{ComputeUpwardRank}(G)$
10: $\text{priority\_queue} \leftarrow \textsf{SortByRank}(V, \text{rank})$
11: $S \leftarrow \emptyset$  *// Scheduled nodes*
12: $\mathcal{L} \leftarrow \textsf{InitializeLockManager}(\mathcal{R})$
13: $\mathcal{TB} \leftarrow \textsf{InitializeTokenBuckets}(\mathcal{R})$
14:
15: **while** $\text{priority\_queue} \neq \emptyset$ **do**
16:     $v \leftarrow \text{priority\_queue.pop}()$
17:     $s_v \leftarrow \textsf{FindEarliestFeasibleSlot}(v, G, S, \mathcal{B}, \mathcal{TB})$
18:     **if** $s_v = \bot$ **then**
19:         **return** error("Unschedulable under constraints")
20:     **end if**
21:     $S \leftarrow S \cup \{(v, s_v)\}$
22: **end while**
23:
24: *// Phase 3: Runtime Coordination Setup*
25: $\text{lock\_order} \leftarrow \textsf{AssignGlobalResourceOrdering}(\mathcal{R})$
26: **return** $(S, \text{lock\_order}, \mathcal{TB})$

---

### E.1. Compilation Pipeline

### E.2. Dependency Construction Subroutines

**Complexity:** $O(|V| \cdot d_{\text{param}})$ where $d_{\text{param}}$ is the maximum number of parameters per node (typically small constant).

**Complexity:**

$$O\left(|V| + |E_{\text{data}}| + \sum_{r \in \mathcal{R}} |V_r|^2\right) = O(|V| + |E_{\text{data}}| + |\mathcal{R}| \cdot |V|^2) \tag{40}$$

In practice, $|V_r| \ll |V|$ for most resources, yielding near-linear behavior.

**Implementation Note:** The conceptual definition above enumerates all pairs in $V_r$, yielding $O(|V_r|^2)$ complexity per resource. In practice, we use a more efficient sweep-line algorithm: sort $V_r$ by topological order once, then maintain the most recent writer and set of active readers in a single linear scan. When encountering a new node $v$: if $v$ writes to $r$, add edges from all active readers and the last writer to $v$, then update the last writer; if $v$ only reads $r$, add an edge from the last writer (if any) to $v$ and add $v$ to active readers. This reduces the per-resource complexity to $O(|V_r| \log |V_r| + |E_r|)$ where $|E_r|$ is the number of resource edges for $r$, making the practical total complexity $O(|V| \log |V| + |E_{\text{data}}| + \sum_r |E_r|)$, which is typically much better than the worst-case quadratic bound.

**Complexity:** $O(|V_r^W| \log |V_r^W|)$ for sorting. Summing over all resources:

$$O\left(\sum_{r \in \mathcal{R}} |V_r^W| \log |V_r^W|\right) \leq O(|\mathcal{R}| \cdot |V| \log |V|) \tag{41}$$

---

**Algorithm 3** Build Data Dependencies

---

1: **Input:** $V, E$
2: $E_{\text{data}} \leftarrow \emptyset$
3: **for all** $v \in V$ **do**
4:     **for all** $p \in \text{dom}(\theta_v)$ **do**
5:         **if** $\theta_v[p] = \texttt{ref}(u, q)$ **then**
6:             $E_{\text{data}} \leftarrow E_{\text{data}} \cup \{(u, v)\}$
7:         **end if**
8:     **end for**
9: **end for**
10: **return** $E_{\text{data}}$

---

**Algorithm 4** Build Resource Dependencies

---

1: **Input:** $V, E_{\text{data}}$
2: $E_{\text{res}} \leftarrow \emptyset$
3: topo_order $\leftarrow$ TopologicalSort$((V, E_{\text{data}}))$
4: **for all** $r \in \mathcal{R}$ **do**
5:     $V_r \leftarrow \{v \in V \mid \exists a : (r, a) \in \rho_v\}$
6:     Order $V_r$ according to topo_order to obtain $\prec_r$
7:     **for all** pairs $(u, v)$ with $u \prec_r v$ **do**
8:         $(r, a_u) \in \rho_u, (r, a_v) \in \rho_v$
9:         **if** $(a_u = \texttt{W} \vee a_v = \texttt{W}) \wedge a_u \neq a_v$ **then**
10:             $E_{\text{res}} \leftarrow E_{\text{res}} \cup \{(u, v)\}$
11:         **end if**
12:     **end for**
13: **end for**
14: **return** $E_{\text{res}}$

---

## E.3. HEFT-Based Scheduling

**Upward Rank Computation.** For node $v$, the upward rank $\text{rank}^u(v)$ measures the length of the longest path from $v$ to any sink:

$$\text{rank}^u(v) = \hat{\tau}_v + \max_{w \in \text{succ}(v)} \text{rank}^u(w) \tag{42}$$

with base case $\text{rank}^u(v) = \hat{\tau}_v$ for sink nodes (nodes with no successors).

**Complexity:** $O(|V| + |E|)$ (linear in graph size).

**Earliest Feasible Slot Finding.** Given node $v$, budget $\mathcal{B}$, and current schedule $S$, we find the earliest time $s_v$ satisfying:

(i) **Dependency constraint:** $s_v \geq \max_{u \in \text{pred}(v)}(s_u + \hat{\tau}_u)$

(ii) **Concurrency constraint:** $\forall t \in [s_v, s_v + \hat{\tau}_v) : |\{w \in S \mid s_w \leq t < s_w + \hat{\tau}_w\}| \leq K_{\max}$

(iii) **Resource lock availability:** For each $(r, a) \in \rho_v$, lock on $r$ can be acquired at $s_v$

(iv) **Rate limit constraint:** For each external resource $r$ with limit $L_r$: $\text{Tokens}_r(s_v) \geq 1$

(v) **Deadline constraint:** $s_v + \hat{\tau}_v \leq T_{\max}$

**Complexity Analysis:** In the worst case, finding a feasible slot requires examining $O(|S|)$ existing scheduled nodes for each constraint check. With event-driven simulation (jumping to next conflict time), typical-case complexity is $O(\log |S|)$ per check via priority queue of event times. Summing over all $|V|$ nodes:

$$O(|V| \cdot |V| \cdot \log |V|) = O(|V|^2 \log |V|) \tag{43}$$

In practice, bucketed timeline representation (discretize time into slots of size $\delta$) reduces this to $O(|V| \cdot T_{\max}/\delta)$.

---

**Algorithm 5** Build Write Chain

---

1: **Input:** $V, r, E_{\text{data}}$
2: $V_r^W \leftarrow \{v \in V \mid (r, \mathtt{W}) \in \rho_v\}$
3: topo_order $\leftarrow$ TopologicalSort$((V, E_{\text{data}}))$
4: Sort $V_r^W$ according to topo_order to get sequence $[v_1, \ldots, v_k]$
5: $E_{\text{sync}}^r \leftarrow \{(v_i, v_{i+1}) \mid i \in [1, k-1]\}$
6: **return** $E_{\text{sync}}^r$

---

**Algorithm 6** Compute Upward Rank

---

1: **Input:** $G = (V, E)$
2: rank $\leftarrow \{\}$
3: topo_order $\leftarrow$ TopologicalSort$(G)$
4: **for all** $v \in$ reverse(topo_order) **do** {Reverse topological order}
5:     **if** succ$(v) = \emptyset$ **then**
6:         rank$[v] \leftarrow \hat{\tau}_v$
7:     **else**
8:         rank$[v] \leftarrow \hat{\tau}_v + \max_{w \in \text{succ}(v)} \text{rank}[w]$
9:     **end if**
10: **end for**
11: **return** rank

---

## E.4. Runtime Resource Coordination

**Global Lock Ordering.** To prevent deadlocks when nodes acquire multiple locks, we assign global total order $\prec_{\text{lock}}$ to resources. We use a stable, deterministic ordering based on the canonical string representation of resource identifiers:

$$r_i \prec_{\text{lock}} r_j \iff \text{canonical}(r_i) < \text{canonical}(r_j) \text{ (lexicographic)} \tag{44}$$

where canonical : $\mathcal{R} \to$ String converts resource identifiers to a normalized string form (e.g., URI with sorted query parameters). This ensures consistency across all executors in distributed settings.

**Token Bucket Rate Limiting.** For resource $r$ with rate limit $L_r$ (requests/second), we maintain token bucket:

$$B_r(t) = \min\left(C_r, B_r(t_0) + L_r \cdot (t - t_0) - N_r(t_0, t)\right) \tag{45}$$

where $C_r$ is bucket capacity, $B_r(t_0)$ is tokens at last update, and $N_r(t_0, t)$ counts requests in interval $[t_0, t]$.

## E.5. Circuit Breaker and Fault Tolerance

**Circuit Breaker State Machine.** For each external resource $r$, we maintain circuit breaker with states $\{\mathtt{closed}, \mathtt{open}, \mathtt{half\text{-}open}\}$:

$$\text{State}(r, t) = \begin{cases} \mathtt{closed} & \text{if } \hat{p}_{\text{fail}}^r(t) < \theta_{\text{fail}} \\ \mathtt{open} & \text{if } \hat{p}_{\text{fail}}^r(t) \geq \theta_{\text{fail}} \wedge t - t_{\text{trip}} < T_{\text{cooldown}} \\ \mathtt{half\text{-}open} & \text{if } t - t_{\text{trip}} \geq T_{\text{cooldown}} \end{cases} \tag{46}$$

where $\hat{p}_{\text{fail}}^r(t)$ is the empirical failure rate in sliding window $[t - W, t]$, $\theta_{\text{fail}}$ is failure threshold, $t_{\text{trip}}$ is time of last circuit open, and $T_{\text{cooldown}}$ is recovery period.

**Saga Pattern for Compensation.** Following the saga model (Garcia-Molina & Salem, 1987), for reversible write operations we generate compensation actions:

$$\text{Compensate}(v) = \begin{cases} \bar{v} & \text{if } v \text{ has reversible semantics} \\ \mathtt{log\text{-}for\text{-}manual\text{-}intervention} & \text{otherwise} \end{cases} \tag{47}$$

where $\bar{v}$ is the inverse operation satisfying:

$$\forall \sigma \in \Sigma : \text{exec}(\bar{v}, \text{exec}(v, \sigma)) \approx_\epsilon \sigma \tag{48}$$

for some small tolerance $\epsilon$ (e.g., eventual consistency delay).

### E.6. Overall Complexity Summary

| Phase | Time Complexity |
|---|---|
| Static verification | $O(|V| + |E|)$ |
| Build $E_{\text{data}}$ | $O(|V| \cdot d_{\text{param}})$ |
| Build $E_{\text{res}}$ (worst) | $O(|V| + |E_{\text{data}}| + |\mathcal{R}| \cdot |V|^2)$ |
| Build $E_{\text{res}}$ (sweep-line) | $O(|V| \log |V| + |E_{\text{data}}| + \sum_r |E_r|)$ |
| Build $E_{\text{sync}}$ | $O(\sum_r |V_r^W| \log |V_r^W|)$ |
| Upward rank | $O(|V| + |E|)$ |
| HEFT scheduling | $O(|V|^2 \log |V|)$ worst-case |
| | $O(|V| \log |V| + |E|)$ typical-case |
| Runtime lock acquire | $O(|\rho_v| \log |\rho_v|)$ per node |
| Token bucket ops | $O(1)$ per request |
| **Total (worst)** | $O(|V|^2 \log |V| + |\mathcal{R}| \cdot |V|^2)$ |
| **Total (typical)** | $O(|V| \log |V| + |E| + \sum_r |E_r|)$ |

*Table 8.* Time complexity of compilation phases. Typical case assumes sparse resource conflicts and event-driven scheduling.

**Space Complexity.** The compiler maintains:

- Dependency graph: $O(|V| + |E|)$

- Schedule: $O(|V|)$

- Lock manager: $O(|\mathcal{R}|)$

- Token buckets: $O(|\mathcal{R}|)$

- Circuit breakers: $O(|\mathcal{R}| \cdot W)$ for sliding windows

Total space: $O(|V| + |E| + |\mathcal{R}| \cdot W)$.

## F. Verification-Gated Skill Evolution: Complete Pipeline

This appendix provides comprehensive details on the verification-gated skill evolution mechanism, expanding on the summary presented in Section 3.5. The pipeline consists of three main stages: candidate extraction, three-stage verification, and staged deployment.

### F.1. Candidate Extraction and Generalization

From successful trajectories $\tau = \{(v_i, x_i, y_i, t_i)\}_{i=1}^{|\tau|}$ where each tuple represents a node execution with its inputs $x_i$, outputs $y_i$, and timestamp $t_i$, we extract reusable patterns through sequential pattern mining on canonicalized Plan IR traces. Each Plan IR DAG is first converted to a linearized sequence via topological sorting with deterministic tie-breaking based on lexicographically smallest node identifiers when multiple nodes have no remaining predecessors. We then apply the PrefixSpan algorithm to the corpus of canonicalized trajectories $\mathcal{D}$, where the support of a pattern $P$ is $\text{supp}(P) = |\{\tau \in \mathcal{D} : P \preceq \tau\}|/|\mathcal{D}|$ and $P \preceq \tau$ indicates that $P$ appears as a contiguous subsequence in $\tau$'s linearized representation. We set a minimum support threshold of 5% to identify frequent patterns, then filter for data flow consistency by verifying that input-output type signatures, parameter references, and resource footprints match across all occurrences.

When multiple pattern instances exhibit structural similarity but differ in specific parameter values or tool selections, we generalize them into parameterized templates via anti-unification. Given two instances $P_1$ and $P_2$, their least general generalization $\text{LGG}(P_1, P_2) = \arg\min_{P: P_1 \trianglelefteq P, P_2 \trianglelefteq P} \text{Cost}(P)$ minimizes generalization complexity $\text{Cost}(P) = \alpha_p \cdot |\text{Params}(P)| + \alpha_w \cdot |\text{Wildcards}(P)| + \alpha_c \cdot |\text{Constraints}(P)|$, balancing introduced parameters, wildcards, and type constraints. During generalization, we synthesize type constraints for template parameters—for example, if a pattern uses different specific tools like `SearchEmail` and `SearchSlack` across instances, we generalize to a parameter constrained to `SearchInterface` if such abstraction exists. We simultaneously generate placeholder pre- and post-conditions derived from the intersection of conditions satisfied by all instances, which are refined during contract verification. After generalization, we verify that templates satisfy $\text{Con}(\pi)$: the DAG remains acyclic under all parameter instantiations, type constraints ensure compatibility, resource footprints remain sound, and idempotency keys are valid for non-pure effects. Templates failing these checks are discarded or retained only as non-executable planning suggestions.

## F.2. Three-Stage Verification

Candidate skills enter a rigorous verification pipeline before library admission. In Stage 1, we synthesize comprehensive test suites $\mathcal{T}_s = \mathcal{T}_{\text{nom}} \cup \mathcal{T}_{\text{bnd}} \cup \mathcal{T}_{\text{err}}$ covering diverse execution scenarios. Nominal test cases $\mathcal{T}_{\text{nom}}$ replay observed inputs from successful execution traces, extracting input parameter values, expected outputs, and execution contexts to form positive test cases that the skill must reproduce. Boundary condition tests $\mathcal{T}_{\text{bnd}}$ explore edge conditions via property-based testing frameworks (Hypothesis, QuickCheck (Claessen & Hughes, 2000)) that automatically generate cases based on parameter type specifications: empty/null values for strings and collections, numeric boundaries from schema constraints including zero and negatives, maximum lengths and cardinalities, and type coercion scenarios at type boundaries. Error mode tests $\mathcal{T}_{\text{err}}$ inject expected failures including network timeouts and connection errors, HTTP 429 rate limiting and 5xx server errors, malformed responses with invalid JSON or missing fields, and resource unavailability simulating database outages and file system errors. For each error mode, we verify graceful recovery via retries or fallbacks, no resource leaks with proper connection and lock cleanup, and appropriate circuit breaker activation for cascading failures. Each test executes in an isolated sandbox with controlled tool execution, passing if outputs match expected values for nominal/boundary cases or if error handling terminates appropriately without crashes or silent failures. Skills must pass at least 95% of nominal tests, 90% of boundary tests, and 100% of critical error mode tests to proceed.

In Stage 2, we verify formal contracts $\forall x \in \text{Dom}(s) : \text{Pre}_s(x) \Rightarrow \text{Post}_s(s(x))$ derived from three sources. Tool schemas provide explicit contracts such as required parameter constraints (e.g., valid email format), output guarantees (e.g., max 100 search results), and side-effect specifications (e.g., unique document IDs). We mine implicit invariants from execution traces using Daikon-style detection (Ernst et al., 2007) to infer output value ranges (e.g., confidence scores in [0,1]), field presence requirements, and relationship invariants (e.g., `output.count == len(output.items)`), validating candidates across all observed traces. For recognized patterns, we instantiate standard contract templates: search patterns verify all results match the query, CRUD patterns check existence pre-conditions and success post-conditions, and aggregation patterns ensure non-empty inputs and non-negative counts. Verification employs two complementary approaches: lightweight randomized testing generates 1000 random inputs satisfying pre-conditions via QuickCheck-style property checking with shrinking to find minimal counterexamples, while symbolic constraint solving encodes contracts as SMT constraints in Z3 (de Moura & Bjørner, 2008) to verify pre-condition satisfiability, search for counterexamples, and prove contract satisfaction for bounded domains within 10-minute timeouts. Skills pass if no counterexamples emerge from randomized testing and symbolic verification either proves correctness or finds no counterexamples within resource limits.

Stage 3 evaluates impact through controlled experiments at two levels. We first test isolated skill correctness by directly invoking it on held-out inputs $\mathcal{I}_{\text{test}}$, computing intrinsic reliability $\text{Acc}_{\text{iso}}(s) = \frac{1}{|\mathcal{I}_{\text{test}}|} \sum_x \mathbf{1}[\text{correct}(s(x), y_{\text{expected}})]$ which must exceed 95%. For end-to-end evaluation, we perform A/B testing on held-out task set $\mathcal{H}$ disjoint from training trajectories, comparing control group execution with current library $\mathcal{S}$ against treatment group with augmented library $\mathcal{S} \cup \{s\}$. To control for planner stochasticity, we use fixed random seeds for LLM sampling, provide identical retrieval candidate sets (control uses a dummy placeholder), and execute under identical environmental conditions with same resource states and simulated latencies. The regression delta $\Delta_{\text{reg}}(s) = \frac{1}{|\mathcal{H}|} \sum_{q \in \mathcal{H}} \left[ \mathbf{1}[\text{fail}(q, \mathcal{S} \cup \{s\}, \xi)] - \mathbf{1}[\text{fail}(q, \mathcal{S}, \xi)] \right]$ measures impact, where positive values indicate regressions. Skills are admitted if $\Delta_{\text{reg}}(s) \leq 0$, functional tests pass, contract verification succeeds, and optionally show meaningful performance improvement $\Delta_{\text{perf}}(s) \geq 0.5\text{s}$ average latency reduction.

## F.3. Staged Deployment and Lifecycle Management

Newly admitted skills progress through a three-phase deployment pipeline ensuring production reliability. In shadow mode, skills appear in planner context and can be selected during planning, but execution always falls back to atomic tool decomposition—the planner generates Plan IR potentially including the new skill, but during compilation the skill is transparently expanded to atomic components while logging what skill-based execution would have done. We collect metrics on planning frequency (how often the planner selects the skill), theoretical performance gain (estimated latency reduction), and counterfactual success (whether the skill-based plan would have succeeded). Shadow mode lasts for minimum 20 task executions or 7 days, allowing passive observation without production risk.

After successful shadow evaluation, skills enter canary deployment where a fraction of compatible traffic routes to actual skill execution. We start with 5% canary traffic and gradually increase to 50% over multiple stages, selecting canary tasks via deterministic hashing on task IDs for reproducibility. Real-time monitoring tracks success rate, latency, retry rate, and error types, triggering automatic rollback if success rate drops by more than 2% compared to control, P95 latency increases by more than 20%, or retry rate increases by more than 50%. Canary deployment continues for minimum 50 task executions or 14 days with sustained performance before promotion to stable status, where skills become available to 100% of compatible traffic, are included in standard library snapshots, and receive continuous monitoring for performance degradation.

Skills exhibiting sustained degradation in stable deployment face demotion: if metrics degrade below threshold for 3 consecutive days, the skill enters warning state with increased monitoring; if degradation persists for 7 days, it demotes to canary with reduced traffic allocation; if metrics remain poor after demotion or critical failures occur, the skill is deprecated and removed from the library. Deprecated skills can only be reintroduced after re-verification through the full pipeline. The skill library maintains version control with semantic versioning (major.minor.patch), where breaking changes like signature modifications increment major version, backward-compatible improvements increment minor version, and bug fixes increment patch version. Clients can pin to specific library versions for reproducibility while production deployments automatically use latest stable versions, with rollback to previous library snapshots supported for emergency recovery.

# G. Detailed Experimental Setup

## G.1. Dataset Details

**StableToolBench** (Guo et al., 2024) provides evaluation on real-world API calls across 49 categories including weather services, e-commerce platforms, social media APIs, and productivity tools. Following the official protocol, we evaluate on the *solvable subset* consisting of 765 queries (filtered from 1,100 total test queries through multi-model solvability voting). The benchmark emphasizes write-heavy workflows and rate-limited services that stress-test semantic effect reasoning and resource-aware compilation. Tasks are executed in a reproducible simulated environment backed by a virtual API server with controlled rate limits and failure injection.

**ShortcutsBench** (Shen et al., 2024) contains 7,627 total queries, with an evaluation set of 5,220 queries evenly distributed across three difficulty levels (1,740 queries per level: easy, medium, hard). The benchmark provides real-world Apple Shortcuts automation workflows with high-quality action sequences requiring both sequential and parallel tool coordination. Each workflow includes ground-truth action graphs with explicit dependencies, parameter bindings, and control flow annotations, enabling evaluation of plan fidelity and structural correctness.

$\tau$**-bench** (Yao et al., 2024) features multi-turn dialogues with domain-specific policy constraints and tool interaction requirements. The benchmark provides two control environments: *retail* domain with 115 test tasks and *airline* domain with 50 test tasks. Tasks stress-test long-term policy compliance, state tracking, and contextual constraint satisfaction across extended interactions. The benchmark proposes pass@k metrics for evaluating consistency and reliability; we primarily report pass@1 rates following standard practice.

**ToolComp** (Nath et al., 2025) contains 485 prompts with 11 tools and 1,731 step-wise labels. The benchmark focuses on tasks with dependent tool chains where each tool's output serves as input for subsequent operations, stressing dependency tracking, data flow analysis, and critical path optimization. Tasks require correct chaining of 3–8 tools with complex input-output mappings.

**TRAJECT-Bench** (He et al., 2025) provides trajectory-level diagnostics across 10 domains with 1,228 total tools. The benchmark includes datasets for different complexity levels: 2,000 simple parallel tasks, 2,000 hard parallel tasks, and 1,870 sequential tasks distributed across domains. Fine-grained error annotations enable attribution analysis across planning errors,

parameter errors, and execution errors, supporting detailed ablation studies of agent components. For our evaluation, we subsample 428 representative trajectories stratified by domain and complexity (using fixed seed 42 for reproducibility) to balance evaluation cost and coverage.

**ToolSeq-500** is a sequential evaluation protocol we construct by uniformly sampling 500 tasks from StableToolBench's training pool (virtual API cache, disjoint from the solvable test split used in Table 1 and from any training trajectories used in skill learning), stratified by category to maintain domain diversity. Task order is randomized using fixed seed 42 for reproducibility. This protocol enables measurement of skill accumulation dynamics, regression rates, and long-horizon learning effects by tracking performance evolution as the agent processes tasks sequentially and updates its skill library.

### G.2. Metric Definitions

**Success Rate (SR)** is computed as the percentage of tasks where the final output satisfies all correctness criteria. For StableToolBench, we use executable checkers that verify API response correctness. For $\tau$-bench, we verify policy compliance via the benchmark's policy checker. For ShortcutsBench, we check both functional correctness and workflow structure.

**End-to-End Latency** measures wall-clock time from task submission to final result delivery, including: (1) planning phase (IR generation for EvoC2F, direct action generation for baselines); (2) compilation phase (only for EvoC2F); (3) tool execution phase with all retries and error handling. We report both mean latency and 95th percentile (P95) to capture tail behavior.

**Retry Rate (RR)** is the average number of retry attempts per task across all tool calls, computed as $\mathrm{RR} = \frac{\sum_{\text{tasks}} \sum_{\text{calls}} \text{retry\_count}}{\#\text{tasks}}$. This metric indicates execution robustness and failure handling quality.

**Tool Calls per Task (TC)** is the average number of tool invocations per task, reflecting planning efficiency and skill reuse. Lower values (when maintaining success rate) indicate better abstraction and skill library utilization.

**Rate-limit Violation Rate (RLV)** is computed from execution logs as $\mathrm{RLV} = \frac{\#\text{rate-limited calls}}{\#\text{total calls}}$, where rate-limited calls are those returning HTTP 429 or equivalent rate-limit errors. This metric validates whether semantic compilation prevents resource conflicts through proper scheduling.

All metrics are reported as the mean over 5 independent runs. Each run uses a different random seed controlling: (1) LLM sampling randomness (different temperature-based samples); (2) task shuffling within evaluation batches; (3) stochastic components in tool execution simulation (network latency jitter, probabilistic failures).

# H. Evaluation Protocols

This appendix details the evaluation protocols and metric definitions for all benchmarks reported in Table 1. We clarify key methodological choices to ensure reproducibility and enable fair comparison with future work.

### H.1. StableToolBench

**Dataset.** We use the StableToolBench evaluation suite (Guo et al., 2024) with 1,100 test queries spanning 49 API categories. Following the official protocol, all evaluations use the *solvable subset* of 765 queries, where solvability is determined through multi-model voting to filter queries with verified executable solutions.

**Metrics and Key Methodological Choices.**

- **Solvable Pass Rate (SoPR):** Following the official StableToolBench protocol, we compute SoPR as the mean score over solvable queries, where each query receives a score based on the evaluator's judgment: `Solved`=1.0, `Unsure`=0.5, `Unsolved`=0.0. **Critical distinction from ToolBench:** StableToolBench addresses evaluation instability in the original ToolBench by: (1) treating `Unsure` cases as 0.5 to provide stable partial credit rather than binary pass/fail assignment, and (2) excluding unsolvable tasks from evaluation to focus on tasks with verified solutions. Our implementation strictly follows this refined protocol.

- **Solvable Win Rate (SoWR):** Proportion of solvable queries where the method outperforms a reference baseline. Following the official win rate computation rule, when one method achieves `Solved` and the other `Unsolved`, the solved method wins; for other cases (both solved, both unsolved, or unsure outcomes), we use GPT-4-Turbo to make win/lose judgments. **Baseline selection:** We use Llama-3.1-70B-Instruct with single-pass execution as the

reference baseline (deviating from the original paper's GPT-3.5-Turbo-0613 baseline) to provide a stronger and more contemporary comparison point reflecting recent model capabilities.

- **Latency (Lat.):** Wall-clock time from query submission to final response, including all tool execution time, network delays, and retry overhead.

- **Tool Calls per Task (TC):** Total number of tool invocations (including retries) averaged over successful completions.

**Evaluator Configuration.** We use GPT-4-Turbo (gpt-4-turbo-2024-04-09) as the evaluator with temperature=0, consistent with StableToolBench's official setup (May 2024 version). All evaluations use the same evaluator version to ensure fair comparison.

## H.2. ShortcutsBench

**Dataset.** We evaluate on the ShortcutsBench evaluation set (Shen et al., 2024) containing 5,220 queries (from 7,627 total) evenly distributed across three difficulty levels: easy, medium, and hard (1,740 queries per level). Each query includes ground-truth action sequences and parameter annotations for Apple Shortcuts workflows.

**Metrics and Key Methodological Choices.** All accuracy metrics are computed over the overall evaluation set (all difficulty levels combined) using the official evaluation scripts:

- **API Selection Accuracy (ASA):** At each prediction step, the agent receives *correct historical actions* from ground truth (teacher-forced evaluation) to isolate API selection errors from cascading failures. ASA measures the fraction of correctly predicted API types across all steps. **Methodological choice:** Following the official protocol, we adopt teacher-forced history where the agent is given correct past actions at each step, rather than using its own predicted history. This design prevents error accumulation and provides a cleaner assessment of API selection capability in isolation.

- **Static Parameter Preset Accuracy ($Acc_{spp}$):** For parameters with static values (e.g., constants, literals), accuracy of correctly filling the ground-truth value. Computed as the ratio of correctly-filled static parameters to total static parameters across all queries.

- **Outputs From Previous Actions Accuracy ($Acc_{ofpa}$):** For parameters that reference outputs from previous actions (using UID+OutputName format), accuracy of correctly identifying and referencing the source action and output field. Following Equation 3 in (Shen et al., 2024), computed as the ratio of correctly identified output references to total required references.

- **Ask-for-Input Recognition Accuracy ($Acc_{afni}$):** Accuracy of correctly identifying parameters requiring external input, including both system-provided inputs (Clipboard, CurrentDate, ExtensionInput) and user-requested inputs (Ask action). Following the official protocol, computed as:

$$Acc_{afni} = \frac{\text{Number of correctly identified Ask-for-Input parameters}}{\text{Total number of Ask-for-Input parameters required}} \quad (49)$$

where the numerator counts all input parameters correctly recognized across all queries and steps, and the denominator represents the total ground-truth Ask-for-Input parameters in the evaluation set.

- **Latency (Lat.):** End-to-end agent loop time. **Note:** ShortcutsBench does not execute real tools; tool outputs are returned from cached ground-truth responses with simulated per-call network delay. Reported latency therefore includes all LLM forward passes (multi-turn for sequential baselines such as ReAct/Toolformer), retrieval and internal reasoning, and the simulated tool-response delays for every invoked action; it excludes only real network round-trips. Temperature=0.2 for all methods.

## H.3. $\tau$-bench

**Dataset.** We evaluate on $\tau$-bench (Yao et al., 2024), which contains multi-turn dialogues with policy constraints across two domains: *retail* (115 test tasks) and *airline* (50 test tasks).

**Metrics and Domain-Specific Reporting.**

- **Domain-Specific Pass Rates ($\text{pass}_{\text{ret}}^1$, $\text{pass}_{\text{air}}^1$):** Following the official $\tau$-bench protocol, we report pass@1 rates *separately* for retail and airline domains rather than aggregating across domains. A task passes if the agent successfully completes all required actions while respecting domain-specific policies (e.g., refund eligibility windows, loyalty program rules). We use the official executable evaluation scripts with ground-truth state validation.

  **Methodological choice:** The two domains have substantially different task distributions and difficulty profiles: retail tasks typically achieve higher pass rates (60–80%) due to simpler policies and more forgiving error margins, while airline tasks are more challenging (35–55%) due to complex multi-step booking policies and stricter compliance requirements. Aggregating these domains into a single metric would obscure domain-specific performance characteristics and hinder reproducibility. We therefore follow $\tau$-bench's official protocol of domain-stratified reporting.

  The benchmark proposes pass@k metrics for evaluating consistency and reliability; we primarily report pass@1 as the standard success rate measure, consistent with common practice in tool-use evaluation.

- **Latency (Lat.):** End-to-end task completion time including all dialogue turns, tool calls, and policy checking overhead. Temperature=0 for deterministic policy compliance.

### H.4. ToolComp

**Dataset.** ToolComp (Nath et al., 2025) provides 485 prompts using 11 tools with 1,731 step-wise labels for dependent tool chains. Each task requires correct chaining of 3–8 tools with ground-truth dependency annotations enabling evaluation of data flow correctness and scheduling quality.

**Metrics.**

- **Success Rate (SR):** Fraction of tasks where the agent produces correct final outputs verified by executable checkers.

- **Dependency Satisfaction Rate (DSR):** Fraction of tool invocations where all input parameters are correctly bound to outputs from upstream dependencies with matching types. Computed by comparing predicted data-flow edges to ground-truth dependency annotations.

- **Critical Path Gap (CPG):** Ratio of actual execution makespan to the theoretical minimum makespan (critical path length assuming infinite parallelism and zero overhead). CPG=1.0 indicates optimal scheduling; higher values indicate suboptimal parallelization or unnecessary serialization.

- **Latency (Lat.):** Wall-clock execution time as in StableToolBench.

### H.5. TRAJECT-Bench

**Dataset.** TRAJECT-Bench (He et al., 2025) provides trajectory-level diagnostics across 10 domains with 1,228 tools. The benchmark includes 2,000 simple parallel tasks, 2,000 hard parallel tasks, and 1,870 sequential tasks distributed across domains, with fine-grained annotations for each step. For our evaluation, we subsample 428 representative trajectories stratified by domain and complexity (using fixed seed 42) to balance evaluation cost while maintaining domain coverage.

**Metrics.** All metrics are step-level accuracies computed over the sampled trajectories:

- **Tool Selection Accuracy:** Fraction of steps where the predicted tool matches the ground-truth tool.

- **Parameterization Accuracy:** Fraction of steps where all tool parameters are correctly filled (correct values for static parameters; correct references for dynamic parameters).

- **Execution Order Accuracy:** Fraction of steps executed in the correct topological order relative to dependencies.

- **Success Rate (SR):** End-to-end task success as in ToolComp.

### H.6. General Experimental Setup

**Concurrency and Runs.** Concurrency limit set to $K_{\max} = 8$ parallel tool executions. All results report the mean over 5 independent runs with different random seeds.

---

**Algorithm 7** Find Earliest Feasible Slot

---

1: **Input:** $v, G, S, \mathcal{B}, \mathcal{TB}$
2: $t_{\min} \leftarrow \max_{u \in \text{pred}(v)}(s_u + \hat{\tau}_u)$  *// Dependency bound*
3: $t_{\text{cand}} \leftarrow t_{\min}$
4: max_iterations $\leftarrow 1000$  *// Safety cap; not a complexity bound*
5: **for** $i = 1$ **to** max_iterations **do**
6:    feasible $\leftarrow$ `true`
7:
8:    *// Check concurrency constraint*
9:    active $\leftarrow \{w \in S \mid [s_w, s_w + \hat{\tau}_w) \cap [t_{\text{cand}}, t_{\text{cand}} + \hat{\tau}_v) \neq \emptyset\}$
10:    **if** $|\text{active}| \geq K_{\max}$ **then**
11:      feasible $\leftarrow$ `false`
12:      $t_{\text{cand}} \leftarrow \min_{w \in \text{active}}(s_w + \hat{\tau}_w)$  *// Jump to next event*
13:      **continue**
14:    **end if**
15:
16:    *// Check resource lock availability (R-R is permitted; flag only W-involving conflicts)*
17:    conflicts $\leftarrow \{w \in S \mid (\exists r : (r, a_v) \in \rho_v \wedge (r, a_w) \in \rho_w \wedge (a_v = \text{W} \vee a_w = \text{W})) \wedge$
18:         $[s_w, s_w + \hat{\tau}_w) \cap [t_{\text{cand}}, t_{\text{cand}} + \hat{\tau}_v) \neq \emptyset\}$
19:    **if** conflicts $\neq \emptyset$ **then**
20:      feasible $\leftarrow$ `false`
21:      $t_{\text{cand}} \leftarrow \min_{w \in \text{conflicts}}(s_w + \hat{\tau}_w)$
22:      **continue**
23:    **end if**
24:
25:    *// Check rate limits*
26:    **for all** $(r, a) \in \rho_v$ where $r$ is external **do**
27:      **if** $\mathcal{TB}[r].\textsf{HasTokens}(t_{\text{cand}}) = $ `false` **then**
28:         feasible $\leftarrow$ `false`
29:         $t_{\text{cand}} \leftarrow \mathcal{TB}[r].\textsf{NextAvailable}(t_{\text{cand}})$
30:         **break**
31:      **end if**
32:    **end for**
33:    **if** $\neg$feasible **then**
34:      **continue**
35:    **end if**
36:
37:    *// Check deadline*
38:    **if** $t_{\text{cand}} + \hat{\tau}_v > T_{\max}$ **then**
39:      **return** $\perp$  *// Unschedulable*
40:    **end if**
41:
42:    **return** $t_{\text{cand}}$  *// Found feasible slot*
43: **end for**
44: **return** $\perp$  *// Timeout after max iterations*

---

---

**Algorithm 8** Acquire Locks

---

1: **Input:** $v, \text{lock\_order}, \mathcal{L}, \text{retry\_count}$  *// retry_count maintained by caller*
2: resources $\leftarrow \{r \mid (r, a) \in \rho_v\}$
3: Sort resources according to $\prec_{\text{lock}}$ to get $[r_1, \ldots, r_k]$
4: acquired $\leftarrow []$
5: timeout $\leftarrow 5.0$ seconds
6: $t_{\text{start}} \leftarrow \textsf{CurrentTime}()$
7: **for** $i = 1$ **to** $k$ **do**
8:     success $\leftarrow \mathcal{L}.\textsf{TryAcquire}(r_i, \max(0, \text{timeout} - (\textsf{CurrentTime}() - t_{\text{start}})))$
9:     **if** $\neg$success **then**
10:        *// Release all acquired locks in reverse order*
11:        **for all** $r \in \textsf{reverse}(\text{acquired})$ **do**
12:            $\mathcal{L}.\textsf{Release}(r)$
13:        **end for**
14:        $\textsf{Sleep}(\min(2^{\text{retry\_count}}, 10)$ seconds$)$  *// Exponential backoff*
15:        **return** `retry`  *// Caller increments retry_count and re-invokes*
16:     **end if**
17:     acquired.$\textsf{append}(r_i)$
18: **end for**
19: **return** `success`

---

**Algorithm 9** Token Bucket Implementation

---

1: **Input:** $r, L_r, C_r$  *// requires $L_r > 0$*
2: tokens $\leftarrow C_r$
3: $t_{\text{last}} \leftarrow \textsf{CurrentTime}()$
4:
5: **function** $\textsf{HasTokens}(t)$:
6:     $\textsf{Refill}(t)$
7:     **return** tokens $\geq 1$
8:
9: **function** $\textsf{Consume}(t)$:
10:     $\textsf{Refill}(t)$
11:     **if** tokens $\geq 1$ **then**
12:         tokens $\leftarrow$ tokens $- 1$
13:         **return** `true`
14:     **else**
15:         **return** `false`
16:     **end if**
17:
18: **function** $\textsf{Refill}(t)$:
19:     $\Delta t \leftarrow t - t_{\text{last}}$
20:     tokens $\leftarrow \min(C_r, \text{tokens} + L_r \cdot \Delta t)$
21:     $t_{\text{last}} \leftarrow t$
22:
23: **function** $\textsf{NextAvailable}(t)$:
24:     $\textsf{Refill}(t)$
25:     **if** tokens $\geq 1$ **then**
26:         **return** $t$
27:     **else**
28:         $\Delta t_{\text{wait}} \leftarrow (1 - \text{tokens})/L_r$
29:         **return** $t + \Delta t_{\text{wait}}$
30:     **end if**

---

---

**Algorithm 10** Circuit Breaker

---

1: **Input:** $r, \theta_{\text{fail}}, W, T_{\text{cooldown}}, n_{\text{min}}$  *// $n_{\text{min}}$: minimum samples before tripping*
2: state $\leftarrow$ `closed`
3: failures $\leftarrow$ deque() *// Timestamps of recent failures*
4: requests $\leftarrow$ deque() *// Timestamps of recent requests*
5: $t_{\text{trip}} \leftarrow \perp$
6:
7: **function** AllowRequest():
8:     $t \leftarrow$ CurrentTime()
9:     CleanWindow($t, W$)
10:
11:     **if** state = `open` **then**
12:         **if** $t - t_{\text{trip}} \geq T_{\text{cooldown}}$ **then**
13:             state $\leftarrow$ `half-open`
14:         **else**
15:             **return** `false`
16:         **end if**
17:     **end if**
18:
19:     **if** state = `half-open` **then**
20:         **return** `true`  *// Allow single probe request*
21:     **end if**
22:
23:     **return** `true`
24:
25: **function** RecordSuccess():
26:     $t \leftarrow$ CurrentTime()
27:     requests.append($t$)
28:     CleanWindow($t, W$)
29:     **if** state = `half-open` **then**
30:         state $\leftarrow$ `closed`
31:         failures.clear()
32:     **end if**
33:
34: **function** RecordFailure():
35:     $t \leftarrow$ CurrentTime()
36:     requests.append($t$)
37:     failures.append($t$)
38:     CleanWindow($t, W$)
39:
40:     $\hat{p}_{\text{fail}} \leftarrow |\text{failures}| / \max(1, |\text{requests}|)$
41:     **if** $|\text{requests}| \geq n_{\text{min}} \wedge \hat{p}_{\text{fail}} \geq \theta_{\text{fail}}$ **then**
42:         state $\leftarrow$ `open`
43:         $t_{\text{trip}} \leftarrow t$
44:     **end if**
45:
46: **function** CleanWindow($t, W$):
47:     **while** $|\text{failures}| > 0 \wedge \text{failures}[0] < t - W$ **do**
48:         failures.popleft()
49:     **end while**
50:     **while** $|\text{requests}| > 0 \wedge \text{requests}[0] < t - W$ **do**
51:         requests.popleft()
52:     **end while**

---

