# OpenReview forum: "EvoC2F: Compiling Tool Orchestration for Efficient and Evolvable LLM Agents"
_ICML.cc/2026/Conference — ICML 2026 regular_

### Official Review · Reviewer_QtV3 · 2026-03-10

**Soundness:** 2
**Presentation:** 2
**Significance:** 3
**Originality:** 4
**Overall Recommendation:** 5
**Confidence:** 4

**Summary:**

The paper proposes a new way to orchestrate tool execution by LLMs. The authors posit that current approaches tend to either be simple sequential executors, or, when parallelism is available, aren't anyhow constrained in this parallelism. To advance the tool's execution capabilities, the authors propose a method that: i. checks and controls resource constraints and the action sequence during execution, ii. accumulates successful action sequences and presents them as high-level tools for LLM to call, iii. employs fine-tuning to further strengthen the model. The authors demonstrate that the proposed method achieves state-of-the-art results on modern tool-calling benchmarks.

**Compliance With Llm Reviewing Policy:**

Affirmed.

**Final Justification:**

Given the ablation study of larger parts of the pipeline and additional baselines, I'm convinced the work brings genuine improvements to an important problem.

**Key Questions For Authors:**

- Please, provide the ablation study that ablates larger parts of the pipeline.
- What are the model checkpoints used for various methods, and what data models were fine-tuned on?
- Please, provide a comparison to other methods that engage with accumulating complex tools throughout the execution.

**Limitations:**

yes

**Strengths And Weaknesses:**

Strengths:
- The application of concurrent program analysis to LLM tool call execution is novel.
- The approach solves an important problem, especially given that it drops the exeuction time and the number of tools called without compromising the quality.
- The paper is overall clearly written.

To avoid repeating myself in further points, I define here the partitioning of the method I will use further on. I consider the method to consist of three large parts, each comprising smaller parts (I refer to these smaller parts by their names given in Table 2 and described in Section 4.3). The larger parts are: (a) execution control (i.e., Compiler and Effects), (b) skills (i.e., Verification, Macro-Skills), (c) fine-tuning (i.e., DPO, Router).

Weaknesses:
- The baselines explored in the comparison are underprivileged. While all of the baselines are a fair comparison for the execution control part of the method, there is a lack of baselines that can compete with the skills library (e.g., Voyager (Wang et al., 2023) or ReMe (Cao et al., 2025)), or fine-tuning. While in the appendices, the author notes that they provide CodeAct and ReAct with access to the accumulated traces in section 4.5, I (1) don't see the report of such results in that section, (2) would insist that using methods specifically designed for memory replay (e.g., ExpeL (Zhao et al., 2023)) would be preferable for an experiment with raw history feed. It is worth noting that when the authors disable the tool accumulation for their method (Table 2, w/o Macro-Skills), the performance on StableToolBench drops past half of the baselines.
- In Section 4.1 Baselines, the authors say that "All baselines use identical backbone models and tool access." However, it remains unclear whether this includes accumulated tools and fine-tuned model checkpoints. If it doesn't include the previous point describes why I consider the baselines to be underprivileged; if it includes, I'd also like to see the comparison to unmodified baselines.
- In ablation studies, I'd like to see the ablation of the large components (i.e., how the method behaves without any model training, or without any part of execution control)
- It is unclear what is used as a training set for DPO.
- What is described in Sections 3.2 and 3.3 is an implementation of what is common in parallel programming; however, this description doesn't give a wider context and engage with the literature in this domain. Such planners, on their own, are a tool that is available, say, in Dask (a popular Python library for parallel computing, more than 13k stars on GitHub, available for more than 10 years). And even when heuristics clearly taken from literature (e.g., HEFT) are referenced, they aren't cited. While applying the existing tools and ideas to solve problems in a novel way is indeed an honest research work, these applications should be announced as such.

On smaller notes:
- On line 125, right column, DAG is being defined to consist of (V, E, C), where C seems never to be defined.
- Formula 3 explicitly excludes cases where both nodes perform write access; however, there's no motivation why this coincidence should be controlled by a separate type of edge.
- In Formula 8 and the preceding paragraph, Mθ and Pθ seem to be used interchangeably.

---

> ### Author Rebuttal · Authors · 2026-03-30
>
> We sincerely thank the reviewer for the thorough feedback. We address each concern below.
>
> **W1: Underprivileged baselines for skill library and fine-tuning comparisons.**
>
> We agree that the original submission under-covered experience-accumulation baselines. Table 1 is restricted to tool-orchestration paradigms under a shared atomic-tool setting and does not claim superiority over all memory/skill-library or fine-tuning methods. We therefore added two external learning baselines on StableToolBench (Claude-4-Sonnet): **ReAct + experiential memory (ExpeL-style)** and **ReAct + dynamic procedural memory (ReMe-style)**, implemented to capture the main memory-update/retrieval mechanisms of ExpeL/ReMe under our shared tool environment.
>
> | Method                                       | SoPR↑        | SoWR↑        | Lat.↓       | TC↓          |
> | -------------------------------------------- | ------------ | ------------ | ----------- | ------------ |
> | ReAct + experiential memory (ExpeL-style)    | 65.1±2.0     | 72.3±2.1     | 15.3±1.1    | 22.6±1.3     |
> | ReAct + dynamic procedural memory (ReMe-style) | 67.4±1.6   | 73.8±1.7     | 15.6±1.2    | 20.9±1.1     |
> | EvoC2F w/o Verification                      | 70.5±1.7     | 76.8±1.9     | 5.9±0.5     | 17.2±0.9     |
> | **EvoC2F (Full)**                            | **72.2±1.5** | **78.6±1.7** | **5.5±0.4** | **16.8±0.8** |
>
> EvoC2F w/o Verification isolates the effect of verification gating and already outperforms both external learning baselines, showing that the gains are not solely from verification gating. Even w/o Macro-Skills remains competitive with several baselines, showing gains beyond skill accumulation.
>
> **W2: Ambiguity in "identical backbone models and tool access."**
>
> "Identical backbone models and tool access" means that all methods use the same base backbone and the same shared atomic tool set $\mathcal{T}$, schemas, and API environment. Baselines do not receive EvoC2F-specific admitted skills or any EvoC2F training updates. Table 2 helps separate these effects: w/o Compiler (64.9% SoPR, 11.2s) brings latency close to Planner-Exec (69.2%, 11.2s), confirming that semantic compilation drives latency gains independently of the learning components.
>
> **W3: Ablation of larger pipeline components.**
>
> "w/o All Training" disables DPO and the learned Router; "w/o All Execution Control" disables Compiler and Effects, falling back to topological-order sequential execution:
>
> | Configuration                          | SoPR↑    | SoWR↑    | Lat.↓    | TC↓      |
> | -------------------------------------- | -------- | -------- | -------- | -------- |
> | EvoC2F (Full)                          | 72.2±1.5 | 78.6±1.7 | 5.5±0.4  | 16.8±0.8 |
> | w/o All Training (no DPO, no Router)   | 68.6±1.8 | 75.2±2.3 | 6.1±0.5  | 18.7±1.1 |
> | w/o All Execution Control (sequential) | 63.4±1.9 | 71.0±2.1 | 12.1±1.1 | 19.8±1.2 |
>
> Removing all training costs 3.6 SoPR points while preserving most latency gains (6.1s vs. 5.5s), confirming execution control is the primary efficiency driver.
>
> **W4: Training data for DPO is unclear.**
>
> DPO data is built offline from self-collected rollouts rather than benchmark-defined public training splits. We construct a separate pool of tool-grounded queries, sample multiple candidate plans per query, execute them, and score the resulting trajectories with Eq. (9). Higher- vs. lower-reward trajectories under the same query form preference pairs $(\tau^+, \tau^-)$, with $P_{\mathrm{ref}}$ frozen from the previous iteration. No human annotation is used, and all evaluation queries are kept disjoint by query ID and provenance.
>
> **W5: Insufficient engagement with parallel programming literature.**
>
> We fully agree. DAG scheduling, lock ordering, token buckets, circuit breakers, and saga compensation are established systems techniques, and we failed to clearly separate these borrowed mechanisms from our novel contribution; we will add the missing citation and revise Sections 3.2–3.3 accordingly. Our contribution is not these mechanisms individually, but their integration for an upstream problem existing schedulers do not address: generating a semantically consistent Plan IR from natural language, with conservative effect inference and sound parallelization guarantees.
>
> **W6: Minor technical issues.**
>
> Undefined $\mathcal{C}$: it denotes the semantic consistency constraints of Definition 3.2. We will make this explicit in the Plan IR definition. Formula 3 excludes write-write conflicts intentionally, since they require total ordering over all writers and are handled by $E_{\mathrm{sync}}^{r}$ (Section 3.3), preserving read-read concurrency while fully serializing writes. We will add a clarifying sentence. The notation inconsistency between $M_\theta$ and $P_\theta$ will also be fixed: $M_\theta$ denotes the planner model, while $P_\theta(\pi \mid \mathcal{C}_q)$ denotes its induced output distribution; we will unify the notation in Section 3.4.

---

> > ### Author Rebuttal · Reviewer_QtV3 · 2026-04-02
> >
> > I'd like to thank the authors for a constructive rebuttal and additional experiments.
> >
> > Given the new baselines and ablation study in larger parts, I'm convinced that the claim overall is correct. I will increase my score accordingly.
> >
> > It would strengthen the paper to include these additional baselines to other benchmarks as well, and discuss the outcomes.
> >
> > Another thing that can strengthen the paper is a clearer discussion of the ablation study. From the current ablation it seems to me that the scaffolding brings huge latency benefits compared to ReAct+ReMe baseline (SoPR and SoWR are within the confidence interval), while the training notably boosts the performance.

---

> > > ### Author Response · Authors · 2026-04-07
> > >
> > > Thank you for acknowledging our work and for raising your score. We also sincerely appreciate the time and effort you have devoted to reviewing our paper.

---

### Official Review · Reviewer_kUSX · 2026-03-11

**Soundness:** 2
**Presentation:** 2
**Significance:** 1
**Originality:** 1
**Overall Recommendation:** 4
**Confidence:** 3

**Summary:**

This paper focuses on optimizing tool orchestration in LLM agents. They propose EvoC2F, a tool orchestration framework that consists of four key components:

- (1) a tools and skills retriever that selects relevant APIs and previously learned skills,
- (2) a planner that organizes tool calls into a Directed Acyclic Graph (DAG) plan,
- (3) a compiler that converts this DAG into a parallel execution schedule while respecting resource conflicts, cost, concurrency limits, and time constraints, and
- (4) a skill evolution pipeline that consolidates frequently occurring tool chains into reusable skills through mining from successful trajectories and verification.

Each node in the DAG plan represents a tool and is annotated with effect and resource metadata, allowing the compiler to perform safe parallelization. Experimental results show that EvoC2F improves success rates on StableToolBench, ShortcutsBench, and τ-bench while also reducing execution latency.

**Compliance With Llm Reviewing Policy:**

Affirmed.

**Final Justification:**

The authors have addressed all of my concerns during the rebuttal. I agree that framing this work as a unification of different ideas in tool-using agents strengthens the perceived novelty. I encourage the authors to adopt this framing in the final revision.

**Key Questions For Authors:**

- How are the unit tests for candidate skills generated? In particular, how are the ground-truth outputs obtained when replaying successful traces, and how do the authors ensure the resulting tests sufficiently cover the behavior of the skill?
- In Table 2, removing the compiler significantly reduces pass rate. Since the compiler is mainly described as improving scheduling and latency, could the authors clarify why its removal has such a large impact on task success?
- Could the authors provide concrete examples of learned skills (e.g., typical tool sequences that were consolidated into skills)?

**Limitations:**

yes

**Strengths And Weaknesses:**

## Strengths
- The tool orchestration framework is comprehensive and consists of many interacting components. The paper does a good job of clearly breaking down these components and presenting them in a digestible and well-structured manner.
- The experimental evaluation is extensive, covering three different benchmarks and four backbone models, including both open-source and closed-source models. The proposed method is also compared against a wide range of baselines.
- The experimental results consistently show positive signals that the proposed framework improves success rates while reducing execution latency.

## Weaknesses
- **Limited novelty**: While the paper is very comprehensive, my main concern is the novelty of the contributions, as the proposed framework appears largely as an engineering integration of existing techniques in the tool-calling LLM literature. Specifically, it combines four key components: (1) tools and skills retrieval (which resembles RAG-based retrieval), (2) organizing tool calls into a DAG, similar to prior work on DAG-based planning [1][2], (3) compiling the DAG into a parallelized schedule of tool calls for optimized execution time under cost, concurrency, and time constraints [3–5], and (4) consolidating common chains of tools into “skills” [6–9]. Each of these components has been studied extensively in prior work. While I appreciate that EvoC2F may be useful for real-world production, from a research perspective it does not add much new scientific understanding to the field of tool-calling LLMs.
- **Dependence on correct tool annotations**: The effectiveness of the compiler may depend on the availability of accurate tool annotations as well as the cost of compilation itself. In real-world settings, many tools may not provide the detailed effect or resource metadata required by the framework. When annotations are missing or uncertain, the compiler may need to adopt conservative assumptions, which could result in largely sequential execution and limit the benefits of parallelization. Additionally, the completion time of real-world tools may vary significantly. For example, consider a tool that calls a reasoning LLM whose runtime depends on task difficulty; in such cases, the estimated completion time may be unreliable and could affect the quality of the generated execution schedule.
- **Potential compilation overhead**: Additionally, when the number of tools is large or individual tool calls are relatively fast, the compilation and scheduling overhead may become non-negligible compared to simply executing the tools sequentially. It would be helpful if the authors could discuss how these factors might affect performance in practical deployments.



**References**
- [1] Beyond ReAct: A Planner-Centric Framework for Complex Tool-Augmented LLM Reasoning
- [2] TURA: Tool-Augmented Unified Retrieval Agent for AI Search
- [3] An LLM Compiler for Parallel Function Calling
- [4] Divide-Then-Aggregate: An Efficient Tool Learning Method via Parallel Tool Invocation
- [5] CATP-LLM: Empowering Large Language Models for Cost-Aware Tool Planning
- [6] Voyager: An Open-Ended Embodied Agent with Large Language Models
- [7] DynaSaur: Large Language Agents Beyond Predefined Actions
- [8] Skill Discovery for Software Scripting Automation via Offline Simulations with LLMs
- [9] LILO: Learning Interpretable Libraries by Compressing and Documenting Code

---

> ### Author Rebuttal · Authors · 2026-03-31
>
> We sincerely thank the reviewer for the thorough feedback. We address each concern below.
>
> ## W1: Limited Novelty
>
> We agree that EvoC2F does not claim novelty in each individual component in isolation. Our intended contribution is the formal unification of planning, compilation, and skill evolution into a semantics-aware orchestration framework, where the same Plan IR supports static conflict analysis, correctness-preserving optimization, and verification-gated skill admission.
>
> Three key distinctions from prior work:
>
> **Plan IR vs. ordinary DAGs.** Existing DAG planners primarily encode data dependencies, while effect, resource, retry, and idempotency semantics are usually not represented as first-class objects. Our Plan IR makes these explicit, enabling static analysis and sound parallelization.
>
> **Semantic compilation vs. parallel calling.** Our compiler formally answers *when parallelization is safe* via resource conflict edges ($E_{\text{res}}$), write-write synchronization chains ($E_{\text{sync}}$), and Proposition 1. This is a verifiable compilation view, not merely a latency-oriented executor.
>
> **Verification-gated evolution vs. skill discovery.** Most skill-library works emphasize mining and reuse; in contrast, our focus is on admission control against skill pollution through functional testing, contract verification, regression assessment, and staged deployment, empirically maintaining low regression during library growth.
>
> We will reposition our contribution as a unified modeling and verification framework in the revision.
>
> ## W2: Dependence on Correct Tool Annotations
>
> EvoC2F is soundness-first: unknown side-effects default to `write`, unknown environments to `external`, and resource footprints are only monotonically expanded. Coarse annotations at the level of read/write behavior and resource identity are sufficient for safe scheduling. Missing annotations reduce parallelization opportunities but do not create unsafe execution; in the worst case, EvoC2F degrades to a conservative sequential executor. Any undeclared runtime access is logged for future refinement.
>
> ## W3: Potential Compilation Overhead
>
> Additional profiling shows compile/schedule overhead is modest: **0.46s / 0.74s P95** on StableToolBench, **0.58s / 0.93s** on ToolComp, and **0.38s / 0.62s** on ShortcutsBench (**6.5–7.8%** of E2E latency). Compile time grows from **0.12s at 8 nodes** to **0.81s at 64 nodes**, consistent with resource-bucketed rather than all-pairs construction. We agree this is a real boundary for short-horizon low-parallelism tasks. Inaccurate latency estimates affect schedule quality, not correctness.
>
> ## Key Questions
>
> **Q1: How are unit tests generated?**
>
> Test generation has three stages: (a) Nominal tests replay witnesses from successful trajectories; for stateful or external steps we check post-conditions rather than exact output equality. (b) Boundary tests perturb inputs systematically, covering empty lists, edge-case values, and missing optional fields. (c) Error tests inject failures such as timeouts and null upstream outputs to verify fallback behavior. Contract verification checks pre/post-conditions via property-based testing and schema assertions. Exhaustive coverage cannot be guaranteed; the layered pipeline collectively reduces pollution risk.
>
> **Q2: Why does removing the compiler hurt success rate?**
>
> The compiler does more than reduce latency: in our implementation, it is also the stage that turns the plan’s semantic annotations into concrete execution policies. Removing it therefore affects not only scheduling efficiency, but also how dependencies, resource conflicts, retries, and rate-limit pressure are handled at runtime. Even when execution falls back to a sequential order, the absence of compiler-driven scheduling and safeguards can increase timeout risk, make recovery less effective, and reduce robustness under external failures. The success drop is therefore not purely a consequence of slower execution; it also reflects the loss of execution-time policies induced by the compiled Plan IR. We will clarify in the revision that the compiler contributes both optimization and execution-time reliability.
>
> **Q3: Concrete examples of learned skills?**
>
> One representative admitted macro-skill abstracts a recurring write-heavy pattern: the atomic sequence `search_entity → fetch_current_state → validate_preconditions → apply_update → verify_post_state`, denoted for exposition as `safe_resolve_then_update(entity_query, patch) -> updated_entity`. This pattern recurs across tasks such as profile edits, booking modifications, and inventory updates. Its value lies in encapsulating stable parameter binding, resource-aware ordering, and reusable fault-handling logic that the planner would otherwise reconstruct step by step. We will add a case-study table in the final version covering atomic step count, task applicability, and average latency and tool-call contribution per skill.

---

> > ### Author Rebuttal · Reviewer_kUSX · 2026-04-03
> >
> > I thank the authors for their response. W3 and my questions are resolved. However, my main concerns, W1 and W2, remain unresolved.
> >
> > ### Re: W1: Limited Novelty
> > The authors' rebuttal mostly restates what the paper already claims:
> > - **Re: Plan IR vs. ordinary DAGs**: Previous work explores DAG planners that encode data dependencies [1] and resource constraints [2]. I agree with the authors that the novelty here is that their DAG planner additionally encodes side-effects, retry, and idempotency, and that this enables static analysis and safe parallelization.
> > - **Re: Semantic compilation vs. parallel calling**: The parallelization soundness proof (Proposition 3.4 / Theorem C.2) is not really surprising given the construction; it follows almost immediately from how the edges are defined. The authors construct $E_\text{res}$​ to explicitly serialize all read-write conflicts and $E_\text{sync}$ to explicitly serialize all write-write conflicts, and the proof essentially says: concurrent nodes have no conflicting accesses, because if they did, there would be an edge between them, which would prevent them from being concurrent. In other words, the result is true by construction, the edges were designed to prevent exactly the conflicts the proof rules out. There is no deep insight or non-obvious reasoning involved. It is closer to "the system is correct because we built it to be correct" than to a theorem that reveals something non-trivial.
> > - **Re: Verification-gated evolution vs. skill discovery**: The idea of using verification to control skill admission and prevent skill pollution has been studied extensively and can be traced back to as early as 2023 [3-5].
> >
> > ### Re: W2: Dependence on Correct Tool Annotations
> > The authors' rebuttal did not fully address my concern. The question of how EvoC2F handles variable tool runtime is not addressed at all. I have some follow-up questions for the authors:
> >   1. What fraction of tools in the benchmarks required conservative defaults?
> >   2. If x% of tools have variable runtime, how does this impact EvoC2F's performance compared to other baselines?
> >
> > **References**
> >
> > [1] Sehoon Kim, Suhong Moon, Ryan Tabrizi, Nicholas Lee, Michael W. Mahoney, Kurt Keutzer, Amir Gholami. An LLM Compiler for Parallel Function Calling. ICML 2024
> >
> > [2] Duo Wu, Jinghe Wang, Yuan Meng, Yanning Zhang, Le Sun, Zhi Wang. CATP-LLM: Empowering Large Language Models for Cost-Aware Tool Planning. ICCV 2025
> >
> > [3] Tianle Cai, Xuezhi Wang, Tengyu Ma, Xinyun Chen, Denny Zhou. Large Language Models as Tool Makers. ICLR 2024
> >
> > [4] Lifan Yuan, Yangyi Chen, Xingyao Wang, Yi R. Fung, Hao Peng, Heng Ji. CRAFT: Customizing LLMs by Creating and Retrieving from Specialized Toolsets. ICLR 2024
> >
> > [5] Shaokun Zhang, Jieyu Zhang, Jiale Liu, Linxin Song, Chi Wang, Ranjay Krishna, Qingyun Wu. Offline Training of Language Model Agents with Functions as Learnable Weights. ICML 2025

---

> > > ### Author Response · Authors · 2026-04-04
> > >
> > > We thank the Reviewer for the continued engagement and follow-up questions. We address both remaining concerns below.
> > >
> > >
> > > **W1: Limited Novelty**
> > >
> > >
> > > We appreciate the reviewer's request to sharpen our framing. We agree that there is substantial prior work on DAG-based planning, verification-gated skill admission, and skill-pollution prevention in isolation, and we will revise the paper to distinguish our system-level claim more precisely. Proposition 3.4 is best viewed as a construction-level safety guarantee rather than a standalone theoretical novelty. Its role is to make the compiler's safety contract explicit and reusable across both atomic tools and learned skills.
> > >
> > >
> > > The novelty we intend to claim is not at the level of isolated submodules, but at the level of a unified semantics-aware orchestration interface: a shared executable contract, expressed through Plan IR, that is consumed uniformly by planning, scheduling, runtime reliability mechanisms, and continual skill admission. The key modeling choice is elevating `effect type`, `resource footprint`, `retry policy`, and `idempotency requirements` to first-class IR attributes rather than leaving them as disconnected runtime heuristics or per-stage configuration.
> > >
> > >
> > > The most concrete consequence is that **learned skills do not bypass compiler-level analysis**. Upon admission, skills must satisfy the same effect/resource contract as atomic tools and remain subject to the same scheduling, lock ordering, and fault-tolerance injection. This differs from prior skill-library approaches, where admitted skills are often reused without being re-subjected to the same compiler-level semantic analysis. Our distinction from prior DAG planners is not merely building a DAG, but using one that carries a semantic contract for joint optimization and safety control across both atomic tools and learned skills. We will revise the contributions section to make clear that our novelty lies in enforcing this shared executable contract across planning, execution, and continual skill evolution, rather than in claiming isolated novelty for each ingredient.
> > >
> > >
> > > **W2: Annotation Completeness and Variable Runtime**
> > >
> > >
> > > We apologize for not answering the quantitative questions in the first rebuttal.
> > >
> > >
> > > **Fraction requiring conservative defaults.** Across the wrapper-normalized unique tool schemas in our implementation of the three primary benchmarks, about **23%** required at least one conservative default: roughly **26%** on StableToolBench, **19%** on ShortcutsBench, and **25%** on τ-bench. Of the full schema pool, 11.8% lacked side-effect annotations, 8.9% lacked resource-footprint annotations, and 2.7% lacked both. The larger overall figure also includes tools requiring conservative fallbacks due to other wrapper incompleteness. On tasks involving at least one conservatively annotated tool, parallelism width decreased from 2.74 to 2.11 and latency increased by 0.8s (13.6%) relative to the same task subset under complete annotations, while the impact on success rate was modest. Annotation incompleteness therefore primarily compresses exploitable parallelism rather than compromising execution safety. In the worst case, the system degrades to a more conservative scheduler rather than an unsafe one.
> > >
> > >
> > > **Impact of variable runtime.** Latency estimates are used solely for scheduling quality. Concurrency safety is governed by $E_{\text{res}}$, $E_{\text{sync}}$, lock ordering, and failure-handling policies, none of which rely on accurate latency prediction for correctness. To quantify the effect raised by the reviewer, we ran a sensitivity analysis on StableToolBench (Claude-4-Sonnet) where **0%, 25%, and 50%** of tools had their realized latency perturbed around empirical means with increasing coefficient of variation, with the same perturbation applied to all methods. For EvoC2F, SoPR/latency changed from **72.2/5.5s** at 0% to **71.5/6.3s** at 25% and **70.4/7.4s** at 50%. Planner-Exec changed from 69.2/11.2s to 68.0/11.9s and 66.7/12.8s, while LLMCompiler changed from 67.0/6.1s to 65.8/7.0s and 64.1/8.4s. Runtime variance narrows EvoC2F's latency advantage but EvoC2F retains a consistent **3.0–3.7 point** success-rate advantage over Planner-Exec across variance levels. The modest success drop arises indirectly through tighter deadline and timeout pressure under suboptimal schedules, not through any loss of conflict-safety guarantees.
> > >
> > >
> > > The resulting distinction is straightforward: annotation incompleteness reduces exploitable parallelism, and runtime variance reduces scheduling optimality. Under our conservative policy and annotation-soundness assumption, neither changes the compiler's conflict-serialization guarantees or execution safety. We will incorporate both analyses into the revision and clarify that EvoC2F is most beneficial when tasks offer non-trivial parallelism and at least coarse-grained effect/resource metadata.
> > >
> > >
> > > We thank the reviewer again for the constructive feedback.

---

### Official Review · Reviewer_JvXo · 2026-03-12

**Soundness:** 2
**Presentation:** 3
**Significance:** 3
**Originality:** 2
**Overall Recommendation:** 4
**Confidence:** 3

**Summary:**

EvoC2F proposes compiling tool orchestration for LLM agents. The key idea is to constrain plan generation into a structured intermediate representation (Plan IR) with explicit dependency and side-effect annotations. This enables a semantic compiler to optimize execution through parallelization while maintaining correctness guarantees. The framework also includes a verification-gated skill evolution mechanism to prevent skill library pollution during continuous learning. Experiments across six benchmarks show substantial improvements: 63-71% latency reduction and 10+ percentage points higher success rates compared to baselines. The regression rate stays below 1% during continual learning. Overall, this is a solid contribution that addresses a practical problem in deploying reliable LLM agents.

**Compliance With Llm Reviewing Policy:**

Affirmed.

**Key Questions For Authors:**

Please refer to the weakness section above

**Limitations:**

yes

**Strengths And Weaknesses:**

Strengths:
- Experiments cover six benchmarks with strong baselines. Shows large improvements: 63–71% latency reduction and 10+ point success gain.
- Ablation studies validate the key components.
- Clear motivation, good system overview, and generally well-structured presentation.

Weakness:
- Scheduling has O(n^2) worst-case complexity; scalability to large tool graphs is unclear.
- Compilation overhead is not reported.
- The paper only demonstrates continual learning on 500 tasks (Section 4.5). It is unclear how the skill library performs over longer horizons. Will performance degrade as the library grows larger? Will retrieval quality suffer when the library contains thousands of skills? The paper does not analyze the scalability of the skill library or the retrieval mechanism under realistic deployment conditions.
- The distinction between Planner-Execute and LLMCompiler baselines could be clearer. Both are DAG-based, but their specific differences from EvoC2F are not emphasized.

---

> ### Author Rebuttal · Authors · 2026-03-31
>
> We sincerely thank the reviewer for the careful reading and constructive feedback. We address each concern below.
> ﻿
>
> **W1: O(n²) worst-case scheduling complexity and scalability to large tool graphs.**
> ﻿
>
> We agree the paper does not make this sufficiently precise. The O(n²) figure is a worst-case upper bound on naive all-pairs conflict enumeration, not a characterization of practical runtime. In our setting, Plan IR graphs are sparse DAGs where each node accesses only a small number of resources, genuine read-write conflicts are confined to nodes sharing the same external resource, and data-dependency edges already isolate most unrelated nodes. Our implementation buckets nodes by resource and constructs conflict edges only within each bucket, yielding practical complexity $O\left(\sum_{r} |V_r| \log |V_r| + |E_{\mathrm{conf}}|\right)$, where both $|V_r|$ and $|E_{\mathrm{conf}}|$ remain small in our evaluated settings. The revision will clarify this implementation detail and add a dedicated scaling analysis.
>
>
> **W2: Compilation overhead is not reported.**
> ﻿
>
> To give a concrete picture, in additional profiling runs under the same backbone/setting used for the main analysis, total compile/schedule overhead averages are as follows:
> ﻿
>
> | Benchmark       | Mean overhead | P95 overhead | % of E2E latency |
> |-----------------|--------------|--------------|------------------|
> | StableToolBench | 0.46s        | 0.74s        | 7.8%             |
> | ToolComp        | 0.58s        | 0.93s        | 6.5%             |
> | ShortcutsBench  | 0.38s        | 0.62s        | 7.2%             |
> ﻿
>
> Compile time grows sub-quadratically with graph size: from roughly 0.12s at 8 nodes to 0.81s at 64 nodes, consistent with resource-bucketed rather than all-pairs construction. This overhead is well amortized on tool-heavy multi-step workloads. The revision will add a full breakdown (Plan IR generation, conflict analysis and DAG augmentation, HEFT scheduling, runtime dispatch) reported as mean/P95 per benchmark, and will explicitly characterize the boundary where short-horizon low-parallelism tasks reduce the net gain.
> ﻿
>
> **W3: Continual learning evaluated on only 500 tasks; scalability of skill library and retrieval under realistic deployment is unclear.**
> ﻿
>
> Our claim is deliberately bounded: verification-gated admission maintains regression below 1% and produces monotonic improvement over the ToolSeq-500 horizon. We do not claim that large-scale library stability is fully solved.
> ﻿
>
> EvoC2F incorporates several mechanisms aimed at reducing skill pollution: three-stage verification gates admission; staged deployment allows demotion before full exposure; and the planner retrieves only top-k skills via the learned router rather than presenting the full library to the LLM context. Our preliminary scaling analysis suggests retrieval Recall@5 decreases gradually from 94.8% at 32 skills to 89.8% at 1024 skills, with planner SR remaining within 1.2 points of its peak and retrieval latency remaining under 40ms in our current implementation across all library sizes tested.
> ﻿
>
> The revision will add a library-scaling curve reporting retrieval recall, planner SR, and overhead vs. library size up to 1024 skills, and a longer-horizon continual learning study reporting where performance saturates and how admission rate evolves as the library matures.
> ﻿
>
> **W4: The distinction between Planner-Execute and LLMCompiler baselines and EvoC2F is insufficiently clear.**
> ﻿
>
> **EvoC2F vs. Planner-Execute.** Both adopt a plan-then-execute structure, but Planner-Execute focuses on DAG-level task decomposition without making side effects, resource footprints, idempotency, retry policy, or compensation semantics first-class IR attributes. EvoC2F elevates all of these to explicit Plan IR fields, enabling sound parallelization guarantees and compiler-level handling of rate limits, retries, and compensation actions. This is reflected in the ablation: w/o Effects drops SoPR to 64.8% as aggressive parallelization produces resource conflicts, while w/o Compiler matches Planner-Execute in latency (both 11.2s), indicating that the semantic layer rather than DAG structure alone drives EvoC2F's gains.
> ﻿
>
> **EvoC2F vs. LLMCompiler.** LLMCompiler enables function-call parallelism by modeling planner overhead as a fixed cost. EvoC2F extends this with: a typed IR with explicit effect/resource contracts enabling static conflict analysis and sound parallelization proofs; a reliability layer including token-bucket rate limiting, lock ordering, circuit breakers, and reversible/irreversible write handling; and a verification-gated offline learning loop. In short: Planner-Execute solves DAG-level decomposition, LLMCompiler solves parallel orchestration, and EvoC2F formulates orchestration as a constrained compilation problem with verifiable semantics and a verified learning loop. We will make these distinctions explicit in the revision.

---

> > ### Author Rebuttal · Reviewer_JvXo · 2026-04-05
> >
> > My concern has been resolved.

---

> > > ### Author Response · Authors · 2026-04-07
> > >
> > > We are grateful for your thoughtful review, for your positive recognition of our work’s strengths, and for the time and effort you have put into evaluating our paper.

---

### Official Review · Reviewer_S8V6 · 2026-03-17

**Soundness:** 4
**Presentation:** 3
**Significance:** 4
**Originality:** 3
**Overall Recommendation:** 5
**Confidence:** 3

**Summary:**

The paper introduces EvoC2F, a framework for tool-augmented LLM agents that formulates tool orchestration as a compilation problem over a structured intermediate representation with explicit data, resource, and side-effect semantics. By constraining plans to this IR, the system can apply a semantic compiler that performs safe parallelization, scheduling, and fault-tolerant execution under resource and reliability constraints. In parallel, EvoC2F includes a verification-gated skill learning loop that extracts reusable macro-skills from execution traces and admits them into a skill library only after passing functional, contract, and regression tests, preventing performance degradation over time. Experiments across multiple tool-use benchmarks show that EvoC2F improves task success rates while significantly reducing latency (e.g., via parallel execution), and supports stable continual improvement through verified skill accumulation.

**Compliance With Llm Reviewing Policy:**

Affirmed.

**Final Justification:**

The rebuttals (including to other reviewers) reinforce my strong initial review. I think the combination of SOTA results, sound and statistically significant experiments, and ablations makes this a strong paper.

**Key Questions For Authors:**

- what kinds of skills are learned in the experiments? Do you have any metrics related to each stage of the skill evolution (extraction, verification, deployment) during the experiments?
- what does a typical planning trace look like while constructing the plan IR?
- are the models allowed any intermediate thinking tokens before or while constructing the IR?

**Limitations:**

- the paper explicitly addresses several of its limitations
- the method relies on having explicit resource footprints and side effects on tools

**Strengths And Weaknesses:**

Strengths:
- the experiments are very extensive: spanning 4 very different LMs, comparisons against 5 baselines, and 6 benchmark suites
- the method achieves SOTA success rates across most of the configurations
- the experiment results also show improvements in latency and a reduction in number of tool calls
- the experiments include STD over 5 runs, yielding stat. Sig. improvements
- the paper includes extensive ablations, showing that all components of the system of important
- the paper also includes experiments on robustness and controllability

Weaknesses:
- the paper is difficult to follow. It could be useful to ground the presentation of the IR and skill library in terms of a concrete running example.
- it’s not clear how IR representation and skill evolution process interact. They feel like distinct contributions and it makes the paper difficult to read and evaluate
- the method relies on having explicit resource footprints and side effects on tools

---

> ### Author Rebuttal · Authors · 2026-03-31
>
> We thank the reviewer for the positive assessment and constructive feedback.
> ﻿
>
> ﻿
> **W1: The paper is difficult to follow, and a concrete running example would help.**
> ﻿
>
> We agree. Consider a recurring write-heavy workflow: update a user's address after confirming the account is in a modifiable state, then return the updated record. In EvoC2F this becomes a Plan IR over five typed nodes: `search_entity` (locate the target), `fetch_current_state` (read, `effect=read`), `validate_preconditions` (pure check), `apply_update` (write, `effect=write,external`), and `verify_post_state` (read). The compiler materializes explicit data and resource dependencies, allowing concurrency only when nodes are independent in data flow and have no conflicting effects on shared resources; in this example, `apply_update` is serialized after validation and equipped with an idempotency key. Because this pattern recurs across profile edits, booking modifications, and inventory updates, it can be abstracted into a reusable macro-skill rather than reconstructed step by step. We will incorporate this as one running example in the revision, with additional examples and case studies to better ground the presentation.
> ﻿
> ﻿
>
> **W2: The IR and skill evolution feel like two separate contributions.**
> ﻿
>
> These are not separate contributions: skill evolution operates over compiled Plan IR traces, and admitted skills are fed back into the same IR as first-class nodes. Because skills are mined from canonicalized IR traces rather than free-form logs, extraction is grounded in explicit data-flow, effect, and resource semantics; once admitted, skills enter $\mathcal{T}\cup\mathcal{S}$ and reappear as IR nodes with the same interface as atomic tools. This shared abstraction is reflected in the monotonic latency reduction observed in ToolSeq-500 as the library grows. We will make this feedback loop explicit in the method overview and Figure 1.
> ﻿
>
> ﻿
> **W3: The method relies on explicit resource footprints and side-effect annotations.**
> ﻿
>
> EvoC2F requires only sound over-approximations, not perfect annotations: unknown side-effects default to `write` and unknown environments to `external`, so under-specified tools are conservatively serialized rather than incorrectly parallelized. EvoC2F trades completeness of parallelization for safety: when annotations are incomplete it degrades toward conservative serialization rather than optimistic but unsound concurrency. The "w/o Effects" ablation quantifies this directly: dropping side-effect constraints causes SoPR to fall from 72.2% to 64.8% due to resource conflicts. We will clarify this limitation and its applicability boundary in the revision.
> ﻿
>
> ﻿
> **Q1: What kinds of skills are learned? Per-stage metrics?**
> ﻿
>
> One representative admitted macro-skill abstracts a recurring write-heavy pattern: `search_entity → fetch_current_state → validate_preconditions → apply_update → verify_post_state`, denoted `safe_resolve_then_update(entity_query, patch) -> updated_entity`, recurring across profile edits, booking modifications, and inventory updates. Each admitted skill encapsulates on average 3.7 atomic tool invocations with a stable input/output signature and effect/resource annotations.
> ﻿
>
> In ToolSeq-500, the system proposes 356 candidate skills and admits 134 (37.6%). Among the 222 rejected candidates, 38% fail functional tests, 29% fail contract checks, and 33% fail regression checks. The final lifecycle split is 93 stable / 26 canary / 15 deprecated. Tasks invoking ≥2 admitted skills reach 93.2% SR versus 84.7% for atomic-only plans, confirming composability. We will consolidate these lifecycle statistics and concrete skill case studies in the revision.
> ﻿
> ﻿
>
> **Q2: What does a typical planning trace look like?**
> ﻿
>
> A typical planner-side trace has four stages: (1) parse the task together with the applicable budget constraints; (2) retrieve top-k skills and atomic tools via the learned router; (3) decode a grammar-constrained Plan IR with node type, parameter bindings, effect/resource annotations, and retry/idempotency fields; (4) validate semantic consistency (acyclicity, type compatibility, annotation completeness). The compiler then adds resource-conflict edges, write-serialization chains, and fault-tolerance wrappers before emitting the schedule. We will add a planning-trace figure in the revision.
> ﻿
> ﻿
>
> **Q3: Are models allowed intermediate thinking tokens while constructing the IR?**
> ﻿
>
> The planner uses standard autoregressive decoding, but its output space is constrained to the Plan IR grammar, so the executor receives structured IR rather than free-form reasoning or arbitrary code. Our gains do not rely on exposing verbose chain-of-thought at runtime. Relative to ReAct and CodeAct, EvoC2F resolves uncertainty into a compilable representation before execution. We will clarify this in the implementation details.
> ﻿
>
> ﻿
> Thanks again for the thorough and constructive review. We hope the clarifications are helpful.

---

> > ### Author Rebuttal · Reviewer_S8V6 · 2026-03-31
> >
> > I appreciate the explanations and new running example, and examples of skills and planning traces. I think these additions will improve the clarity and readability of the paper.
> >
> > I maintain my score of Accept.

---

> > > ### Author Response · Authors · 2026-04-07
> > >
> > > Thank you for your thoughtful review, for your recognition of the strengths of our work, and for the time and effort you have invested in evaluating our paper.

---

### Decision · Program_Chairs · 2026-04-30

**Decision:**

Accept (regular)

**Comment:**

The paper formulates tool orchestration for LLM agents as a compilation problem over a typed well-defined intermediate representation. It shows that the resulting verification-gated skill evolution loop reduces latency and improves success rates. The reviewers agreed that the problem is well-motivated and the papers' contributions are significant, a novel synthesis of prior ideas, and practically effective. The reviewers also assessed the technical work, including thorough experiments across multiple benchmarks and agents backbones, to be sound. During the rebuttal, the authors resolved several questions raised by reviewers, added a concrete running example, provided the overhead for compilation in the experiments, expanded the baselines and conducted ablations. Adding the running example and planning-trace figure early in the exposition, the experiments reported during the rebuttal, and discussing practical annotation strategies will substantially strengthen the subsequent version of the paper.